# Data Reconstruction: Identifiability and Optimization with Sample Splitting

**Yujie Shen** [1]  **Zihan Wang** [2]  **Jian Qian** [3]  **Qi Lei** [2 4]

## Abstract

Training data reconstruction from KKT conditions has shown striking empirical success, yet it remains unclear when the resulting KKT equations have unique solutions and, even in identifiable regimes, how to reliably recover solutions by optimization. This work hereby focuses on these two complementary questions: identifiability and optimization. On the identifiability side, we discuss the sufficient conditions for KKT system of two-layer networks with polynomial activations to uniquely determine the training data, providing a theoretical explanation of when and why reconstruction is possible. On the optimization side, we introduce sample splitting, a curvature-aware refinement step applicable to general reconstruction objectives (not limited to KKT-based formulations): it creates additional descent directions to escape poor stationary points and refine solutions. Experiments demonstrate that augmenting several existing reconstruction methods with sample splitting consistently improves reconstruction performance. Code is available at https://github.com/mukerr/data_recon.

## 1. Introduction

Deep neural networks (DNNs) have achieved remarkable success across a wide range of tasks, from image classification to natural language processing. Despite their impressive generalization, DNNs are known to memorize training data (Zhang et al., 2021; Feldman, 2020; Feldman & Zhang, 2020). Understanding how training data is encoded in network parameters is therefore of both theoretical and practical importance, with direct implications for general-ization, robustness, and data privacy (Carlini et al., 2021; Song et al., 2017), as sensitive information could potentially be reconstructed from a trained model.

A natural approach to study memorization is dataset reconstruction: recovering the original training samples solely from the parameters of a trained network. Recent works have shown that such reconstruction is possible for homogeneous neural networks trained with gradient-based methods (Haim et al., 2022; Buzaglo et al., 2024; Loo et al., 2024). Haim et al. (2022) leverage the theory of implicit bias (Lyu & Li, 2020; Ji & Telgarsky, 2020) that homogeneous networks trained with gradient flow converge to solutions satisfying the KKT conditions of a maximum-margin problem. This insight provides a principled framework for training-data reconstruction: if the system of KKT equations can be inverted, one can recover information about the original training samples. Despite encouraging empirical success, two fundamental and closely interconnected challenges remain:

*Question 1. Identifiability: When do the KKT equations uniquely determine the training samples, such that the training samples are theoretically recoverable?*

In general, the KKT equations may admit multiple solutions, and reconstruction could be merely heuristic, data-dependent, or unreliable. In this paper, we rigorously analyze two-layer networks with polynomial activations and prove that with cubic or higher-degree activations, margin samples are exactly recoverable for moderately wide networks.

*Question 2. Optimization: What algorithmic strategies can effectively optimize the data reconstruction objective that is high-dimensional and nonconvex?*

Even when the data are identifiable, solving the nonconvex inverse problem remains nontrivial and optimization is crucial for realizing the theoretical potential of reconstruction. Existing approaches fix an oversized candidate set and apply standard gradient descent. In this paper, we introduce a sample splitting algorithm to help escape plateaus where gradient-based methods may stagnate. The idea is conceptually inspired by neuron splitting techniques proposed for neural architecture optimization (Liu et al., 2019).

Our contributions could be summarized as follows:

[1]Zhili College, Tsinghua University, Beijing, China [2]Courant Institute for Mathematical Sciences, New York University, New York, USA [3]School of Computing and Data Science, University of Hong Kong, Hong Kong, China [4]Center for Data Science, New York University, New York, USA. Correspondence to: Yujie Shen <shenyj22@mails.tsinghua.edu.cn>.

*Proceedings of the 43rd International Conference on Machine Learning*, Seoul, South Korea. PMLR 306, 2026. Copyright 2026 by the author(s).

- We establish identifiability results that for two-layer networks with polynomial activations of degree three or higher, training samples can be exactly recovered from the KKT equations under moderate width conditions.

- We propose sample splitting as a flexible optimization strategy to improve reconstruction quality, especially in nonconvex settings, by introducing additional descent directions.

- We empirically demonstrate that sample splitting improves reconstruction quality across several existing reconstruction methods.

## 2. Related Work

**Privacy attacks in deep learning.** Prior work has shown that trained machine learning models may leak sensitive information about their training data in various forms, giving rise to several classes of privacy attacks. *Membership inference* attacks determine whether a specific data point was included in the training set (Shokri et al., 2017; Carlini et al., 2021), while *model inversion* attacks aim to recover sensitive attributes or class representatives from trained models (Fredrikson et al., 2015; Yang et al., 2019). In distributed settings, such as collaborative and federated learning, these risks are further amplified (He et al., 2019). *Gradient inversion* attacks recover information about training data by matching model gradients (Zhu et al., 2019), with subsequent improvements leveraging label leakage (Zhao et al., 2020), hand-crafted regularizers, or strong image priors (Geiping et al., 2020). These attacks typically focus on reconstructing individual samples or partial information.

**Dataset Reconstruction.** Beyond single-sample or attribute-level leakage, a more severe threat is *dataset reconstruction*, which aims to recover a large fraction or even the entirety of the training set using only learned model parameters. Haim et al. (2022) first demonstrated that for homogeneous binary classifiers, many training samples can be reconstructed by exploiting the KKT conditions satisfied at convergence (Lyu & Li, 2020; Ji & Telgarsky, 2020). This line of work is later extended to the multi-class setting and to larger scale pretrained models by Buzaglo et al. (2024); Oz et al. (2024). Loo et al. (2024) show that for networks operating in the neural tangent kernel (NTK) regime (Jacot et al., 2018), the entire training dataset can be provably reconstructed, assuming access to the full parameter initialization. To relax this assumption and handle more practical deep neural networks with nonlinear training dynamics, Tian et al. (2025) propose simulating training dynamics to enable dataset reconstruction. Concurrently, Iurada et al. (2025) establish identifiability conditions for random features models, and Refael et al. (2025) prove that the original KKT-based objective suffers from exponential local minimum. Despite these advances, a precise characterization of when KKT-based reconstruction uniquely determines training samples remains open, and our work addresses this gap.

**Optimization and escaping saddle points.** Our work also relates to the broader literature on nonconvex optimization, particularly studies on escaping saddle points. Classical results show that stochastic gradient methods can avoid strict saddles through random initialization or injected noise (Ge et al., 2015; Jin et al., 2017). A complementary line of research seeks to improve optimization behavior by modifying model parameterizations or architectures, such as neural splitting methods that escape parametric local optima by progressively augmenting the network (Liu et al., 2019). While these approaches focus on altering the optimization dynamics via stochasticity or model design, our work is conceptually related but operates at a different level. We consider data-level splitting strategies that reshape the reconstruction landscape itself. This perspective provides a new mechanism for improving optimization in reconstruction problems.

## 3. Problem Setup

**Setup.** We consider the problem of reconstructing training data from the parameters of a trained neural network. Let $\boldsymbol{\theta} \in \mathbb{R}^P$ denote the parameters of a model $\Phi(\boldsymbol{\theta}; \cdot) : \mathbb{R}^d \to \mathbb{R}$ trained on a dataset $\{(x_i, y_i)\}_{i=1}^n$ using a loss function $\ell : \mathbb{R} \to \mathbb{R}$. The empirical loss is given by $\mathcal{L}(\theta) := \sum_{i=1}^n \ell(y_i, \Phi(\theta; x_i))$. Dataset reconstruction aims to recover the training samples $\{(x_i, y_i)\}_{i=1}^n$ given access only to the trained parameters $\boldsymbol{\theta}$.

**KKT-Based Reconstruction Method.** Haim et al. (2022) propose a reconstruction approach based on the implicit bias of gradient descent. For binary classification with logistic loss and homogeneous models, gradient flow is known to converge in direction to a Karush–Kuhn–Tucker (KKT) point of the maximum-margin problem

$$\min_{\boldsymbol{\theta}} \frac{1}{2}\|\boldsymbol{\theta}\|^2 \quad s.t. \ y_i\Phi(\boldsymbol{\theta}; \boldsymbol{x}_i) \geq 1, \forall i \in [n].$$

A KKT point $\tilde{\boldsymbol{\theta}}$ satisfies the following KKT conditions: there exist $\lambda_1, ..., \lambda_n \in \mathbb{R}$ such that

$$\tilde{\theta} = \sum_{i=1}^n \lambda_i y_i \nabla_\theta \Phi(\tilde{\theta}; x_i) \text{ (stationarity)}$$

$y_i\Phi(\tilde{\theta}; x_i) \geq 1, \forall i \in [n]$ (primal feasibility)

$\lambda_1, \ldots, \lambda_n \geq 0$ (dual feasibility)

$\lambda_i = 0$ if $y_i\Phi(\tilde{\theta}; x_i) \neq 1, \forall i \in [n]$ (complementary slackness)

Motivated by these conditions, Haim et al. (2022) propose to recover training data by optimizing over candidate samples $\{(x_i, y_i)\}_{i=1}^k$ and multipliers $\{\lambda_i\}_{i=1}^k$ with $k \geq 2n$,

minimizing the reconstruction objective

$$L_{total}(\{x_i\}_{i=1}^k, \{\lambda_i\}_{i=1}^k) = \alpha_1 \|\boldsymbol{\theta} - \sum_{i=1}^k \lambda_i y_i \nabla_\theta \Phi(\boldsymbol{\theta}; x_i)\|^2$$

$$+ \alpha_2 \sum_{i=1}^k \max(-\lambda_i, 0) + \alpha_3 L_{prior}$$

where $\alpha_1, \alpha_2, \alpha_3 \in \mathbb{R}$ are tunable hyperparameters of the different losses, and $L_{prior}$ represents some simple dataset constraints such as bounded pixel values.

This formulation has been shown empirically to recover a large fraction of training samples. However, the theoretical foundations of KKT-based reconstruction remain incomplete. Since the optimal value of reconstruction loss is zero by construction, reconstruction is equivalent to solving

$$\boldsymbol{\theta} = \sum_{i=1}^k \hat{\lambda}_i y_i \nabla_\theta \Phi(\boldsymbol{\theta}; \hat{x}_i)$$

which corresponds to $P$ equations with $k(d+1)$ unknowns. When $P < k(d+1)$, the system is under-determined, making recovery impossible in general. This is consistent with prior empirical observations that models trained on fewer samples are more vulnerable to reconstruction in terms of both quantity and quality (Buzaglo et al., 2024; Haim et al., 2022).

On the other hand, even when $P > k(d+1)$, the system is over-determined and highly nonconvex. In this regime, the reconstruction objective $L_{total}$ can in principle admit multiple global minima (all achieving value zero), and minimizing it may not recover the true training data. This raises a fundamental theoretical question: *why should minimizing $L_{total}$ recover the true data at all?*

The answer lies in *identifiability*: if the true training samples are the *unique* set of data points satisfying the KKT equations, they constitute the unique zero of $L_{total}$. In this case, any global minimizer of the reconstruction objective is guaranteed to recover the true data—making the empirical success of these methods theoretically principled rather than heuristic. Section 4 formalizes this reasoning by establishing sufficient conditions under which such uniqueness holds for two-layer networks with polynomial activations.

From an optimization perspective, the over-determined structure also leads to optimization challenges and sensitivity to hyperparameters, explaining the empirical difficulty of stable reconstruction. This motivates the improvement of optimization strategies, which we introduce in Section 5.

# 4. Identifiability of Two-Layer Networks

In this section, we analyze when training data are identifiable from the KKT conditions for two-layer networks. We start from the simplest case, a linear activation, and find that identifiability fails. We then move to nonlinear networks with polynomial activations. Polynomial activations provide a tractable yet expressive family of homogeneous nonlinearities, allowing us to study how increasing the activation degree changes the structure of the KKT system and, consequently, the identifiability of the input samples.

## 4.1. Linear networks

Consider a two-layer linear network of the form $\Phi = a^T W x \quad a \in \mathbb{R}^m, W \in \mathbb{R}^{m \times d}, x \in \mathbb{R}^d$. Under the KKT stationarity condition, we have $\theta = \sum_{i=1}^n \lambda_i y_i \nabla_\theta \Phi(\theta; x_i)$, which decomposes as

$$a = \sum_{i=1}^n \lambda_i y_i W x_i, \quad W_j = \sum_{i=1}^n \lambda_i y_i a_j x_i, \quad j = 1, \cdots, m$$

Collecting the equations into matrix form gives

$$\begin{bmatrix} W \\ a_1 I_d \\ \vdots \\ a_m I_d \end{bmatrix}_{(m+1)d \times d} \cdot (\sum_{i=1}^n \lambda_i x_i y_i)_{d \times 1} = \begin{bmatrix} a \\ w_1 \\ \vdots \\ w_m \end{bmatrix}_{m(d+1) \times 1}$$

Therefore, the KKT system determines only the aggregated quantity $\sum_{i=1}^n \lambda_i x_i y_i$, but not the individual samples $x_i$. Hence a linear two-layer network cannot yield identifiable training inputs.

## 4.2. Homogeneous polynomial activations

We now extend the analysis from linear to nonlinear activations. To maintain homogeneity—an essential condition for the implicit-bias and KKT theory—we focus on pure-power activations of the form $\sigma(t) = t^\alpha$, which form a rather simple class of nonlinear homogeneous functions. This allows us to isolate the effect of activation degree $\alpha$ on identifiability.

Consider $\Phi = a^T \sigma(Wx) = \sum_{j=1}^m a_j(W_j^\top x)^\alpha \quad a \in \mathbb{R}^m, W \in \mathbb{R}^{m \times d}, x \in \mathbb{R}^d$ where $\sigma(x) = x^\alpha$. Then, according to the KKT condition, we have

$$a_j = \sum_{i=1}^n \lambda_i y_i(W_j^\top x_i)^\alpha, \quad W_j = \alpha \sum_{i=1}^n \lambda_i y_i a_j(W_j^\top x_i)^{\alpha-1} x_i$$

for $j = 1, \ldots, m$.

**Quadratic activation.** When the activation is quadratic ($\alpha = 2$), the above reduces to

$$a_j = W_j^\top(\sum_{i=1}^n \lambda_i y_i x_i x_i^\top)W_j, \quad W_j = 2a_j(\sum_{i=1}^n \lambda_i y_i x_i x_i^\top)W_j.$$

Thus, all KKT equations depend only on the second-order moment matrix $M = \sum_{i=1}^{n} \lambda_i y_i x_i x_i^\top$. Consequently, the KKT system identifies only the moment matrix $M$, not the individual samples $x_i$.

**Polynomial activation** ($\alpha \geq 3$). For $\alpha \geq 3$, the same KKT conditions implicitly determine a higher-order symmetric tensor. This tensor can be identified from the trained neuron parameters by polynomial interpolation. Then, for almost all cases, the tensor admits a unique orthogonal decomposition, yielding reconstruction of all active samples. Here we define the active set by $S := \{i : \lambda_i \neq 0\}$, aligning the goal in Haim et al. (2022) to reconstruct all samples on the margin. The detailed analysis is presented in Appendix A.

Specifically, let $b_i := \lambda_i y_i$ and define the order-$\alpha$ symmetric tensor

$$\mathcal{T} := \sum_{i=1}^{n} b_i\, x_i^{\otimes \alpha}.$$

We also use its contraction map

$$f(w) := \mathcal{T}(I, w, \ldots, w) = \sum_{i=1}^{n} b_i (w^\top x_i)^{\alpha-1} x_i,$$

which is a homogeneous polynomial map of degree $\alpha - 1$. We introduce $f$ because the KKT equations of the network $\Phi(x) = \sum_{j=1}^{m} a_j (W_j^\top x)^\alpha$ provide direct evaluations of $f$:

$$\alpha a_j f(W_j) = W_j, \qquad j = 1, \ldots, m. \tag{1}$$

To turn these evaluations into identification, we view $f$ as an unknown element of a finite-dimensional linear space. Let

$$N := \binom{d + \alpha - 2}{\alpha - 1}$$

be the dimension of the space of homogeneous polynomials of degree $\alpha - 1$ in $d$ variables. Concretely, if $\varphi_{\alpha-1}(w) \in \mathbb{R}^N$ denotes the vector of all monomials of total degree $\alpha - 1$, then there exists $A \in \mathbb{R}^{d \times N}$ such that $f(w) = A\varphi_{\alpha-1}(w)$. Here we kernelize this interpolation using the degree-$(\alpha - 1)$ polynomial kernel $\kappa(u, v) = (u^\top v)^{\alpha-1}$, leading to the Gram matrix

$$K \in \mathbb{R}^{m \times m}, \qquad K_{pq} := (W_p^\top W_q)^{\alpha-1}.$$

A full-rank condition on $K$ is precisely what ensures the interpolation problem has a unique solution.

**Theorem 4.1.** *Assume $\alpha \geq 3$ and let $(a, W)$ be any KKT point associated with samples $\{(x_i, y_i)\}_{i=1}^{n}$ and multipliers $\{\lambda_i\}_{i=1}^{n}$. Suppose the interpolation condition holds:*

$$\text{rank}(K) = N, \tag{2}$$

*Then $\mathcal{T}$ is uniquely determined by $(a, W)$. Moreover, if we assume $\{x_i\}_{i=1}^{n}$ are independent and satisfies $\|x_i\|_2 = 1$, one can recover the active components $\{(x_i, b_i)\}_{i \in S}$ from $\mathcal{T}$ up to permutation almost surely.*

*Proof sketch. Step 1.* Equation (1) provides $m$ evaluations of the degree-$(\alpha - 1)$ polynomial map $f$. Under (2), the induced interpolation linear system has a unique solution, which can be expressed in kernel form using $K^\dagger$. This pins down $f$, and therefore uniquely pins down $\mathcal{T}$.

*Step 2.* Once $f$ is identified, we can compute its Jacobian $Df(v)$ with a Jennrich-style algorithm (Leurgans et al., 1993). For $\alpha \geq 3$, $Df(v)$ corresponds to a weighted second-moment operator of the form

$$\sum_{i \in S} b_i (x_i^\top v)^{\alpha-2} x_i x_i^\top,$$

so choosing $v$ provides a data-dependent reweighting of the components. For almost every $v$, this operator is invertible on the signal subspace, and thus induces a transform $Q(v)$ that maps the components to an orthonormal set. Applying the same transform to $\mathcal{T}$ yields an orthogonally decomposable symmetric order-$\alpha$ tensor. For almost every $v$, the resulting orthogonal decomposition is unique, so mapping back through $Q(v)$ ensures that we recover $\{x_i\}_{i \in S}$. $\square$

**Remark 4.2.** In Theorem 4.1, it is required that $K \in \mathbb{R}^{m \times m}$ be a matrix with $\text{rank}(K) = N$, indicating the width of the network $m \geq N$. It can grow rapidly when the input dimension $d$ and the polynomial degree $\alpha$ are relatively large. However, this dependency is tied to the exact-recovery analysis and it is consistent with the overparameterized regime of modern deep networks.

### 4.3. Non-homogeneous polynomial activations

In Section 4.2, we focused on the homogeneous activation $\sigma(t) = t^\alpha$, where the implicit-bias limit admits a clean KKT characterization and leads to identifiability. We now explain why the same conclusion extends to *non-homogeneous* polynomial activations.

**Homogenization.** Consider a two-layer network

$$\Phi_\sigma(\theta; x) = \sum_{j=1}^{m} a_j \sigma(w_j^\top x), \qquad \theta = (a, W),$$

with a polynomial activation

$$\sigma(t) = \sum_{k=0}^{\alpha} c_k t^k, \qquad c_\alpha \neq 0, \ \alpha \geq 3.$$

Following Cai et al. (2025), define the homogenization of the model by

$$\Phi_{\mathrm{H}}(\theta; x) := \lim_{r \to \infty} \frac{\Phi_\sigma(r\theta; x)}{r^M}, \tag{3}$$

where $M$ is the smallest degree for which the limit exists and is nonzero. For polynomial $\sigma$, $\Phi_{\mathrm{H}}$ is simply the model obtained by replacing $\sigma$ with its top-degree homogeneous part $\sigma_{\mathrm{H}}(t) := c_\alpha t^\alpha$, namely

$$\Phi_{\mathrm{H}}(\theta; x) = \Phi_{\sigma_{\mathrm{H}}}(\theta; x) = c_\alpha \sum_{j=1}^{m} a_j \, (w_j^\top x)^\alpha. \quad (4)$$

**Implicit bias and reconstruction.** The main message of Cai et al. (2025) is that, under strong separability and their near-homogeneity assumptions, gradient flow on $\Phi_\sigma$ drives $\|\theta_t\| \to \infty$ while the direction converges, and the limiting direction satisfies the KKT conditions of a max-margin problem induced by the homogenized model $\Phi_{\mathrm{H}}$ in (3). Combining this with (4) shows that, for polynomial activations, the asymptotic KKT system is exactly the one corresponding to the homogeneous $\alpha$-power network with activation $\sigma_{\mathrm{H}}(t) = c_\alpha t^\alpha$, which is the setting we discussed above.

Similar to Section 4.2, only active constraints appear in the KKT stationarity equations. Let $\{\lambda_i^{\mathrm{H}}\}$ denote the KKT multipliers of the max-margin problem associated with $\Phi_{\mathrm{H}}$, and define the active set

$$\mathcal{S}_{\mathrm{H}} := \{ \, i \in [n] : \lambda_i^{\mathrm{H}} > 0 \, \}.$$

Our reconstruction results therefore apply to the samples $\{x_i\}_{i \in \mathcal{S}_{\mathrm{H}}}$. In particular, whenever the neuron-rank condition in Section 4.2 holds, the same tensor-based identifiability and reconstruction conclusions remain. See details in Appendix A.3.

# 5. The Sample Splitting Algorithm

In this section, we address the optimization issue of data reconstruction, where we introduce a splitting-based optimization method. The proposed algorithm is motivated by two fundamental challenges shared by existing reconstruction approaches: (i) the reconstruction objective is highly nonconvex and often contains large flat plateaus where standard gradient descent stagnates, and (ii) the number of training samples is unknown. Splitting creates new candidate samples by perturbing existing ones along directions of negative curvature. This allows us to escape plateaus, refine ambiguous candidates, and adaptively change the number of reconstructed samples.

## 5.1. Beyond KKT: a unified reconstruction objective

Although our identifiability analysis focuses on KKT-based reconstruction, the resulting optimization problem admits a more general interpretation. Several recent reconstruction methods, despite being derived under different theoretical settings, lead to objectives of a common form. In particular,

we consider reconstruction objectives of the form

$$L(\{x_i\}_{i=1}^k, \{\lambda_i\}_{i=1}^k) = \|\theta - \sum_{i=1}^{k} \lambda_i f(\theta; x_i)\|^2 \quad (5)$$

where $x_i \in \mathbb{R}^d$ denote reconstructed samples, $\lambda_i \in \mathbb{R}$ are their associated weights, and $f(\cdot)$ denotes a generic reconstruction map (with specific choices described below). We omit regularization terms and focus on the sample-dependent component, which is the primary source of nonconvexity.

The formulation subsumes several representative reconstruction methods through different choices of $f(\cdot)$:

- **KKT-based reconstruction** (Haim et al., 2022): $f(\boldsymbol{\theta}; x_i) = y_i \nabla_\theta \Phi(\boldsymbol{\theta}, x_i)$;

- **Multiclass extension** (Buzaglo et al., 2024): $f(\boldsymbol{\theta}; x_i) = \nabla_\theta \big[ \Phi_{y_i}(\boldsymbol{\theta}, x_i) - \max_{j \neq y_i} \Phi_j(\boldsymbol{\theta}, x_i) \big]$;

- **NTK-based attack** (Loo et al., 2024): $f(\boldsymbol{\theta}; x_i) = \nabla_\theta \Phi_{\theta_0}(x_i)$.

## 5.2. Algorithm

Motivated by the nonconvexity of the reconstruction objective, we propose a curvature-aware optimization method. The key idea is to locally refine reconstructed samples by splitting them along directions of negative curvature, thereby escaping flat regions.

**Two-phase optimization.** The proposed method alternates between two complementary phases:

- *Phase I (first-order descent):* jointly optimize $(x, \lambda)$ using gradient-based updates.

- *Phase II (sample splitting):* detect directions of negative curvature and refine reconstructed samples by splitting.

**Sample splitting.** To enable adaptive refinement, we allow each reconstructed sample $x_i$ to be replaced by $k_i$ offsprings

$$\boldsymbol{x}_i := \{x_i^{[j]}\}_{j=1}^{k_i}, \ \boldsymbol{\lambda}_i := \{\lambda_i^{[j]}\}_{j=1}^{k_i},$$

$$\text{s.t. } \sum_{j=1}^{k_i} \lambda_i^{[j]} = \lambda_i, \ \lambda_i^{[j]}/\lambda_i > 0.$$

The corresponding augmented objective becomes

$$L(\{\boldsymbol{x}_i\}_{i=1}^k, \{\boldsymbol{\lambda}_i\}_{i=1}^k) = \|\boldsymbol{\theta} - \sum_{i=1}^{k} \sum_{j=1}^{k_i} \lambda_i^{[j]} f(\boldsymbol{\theta}; x_i^{[j]})\|^2$$

This construction preserves the objective value when all off-springs coincide with the original sample.

We characterize the local curvature relevant to splitting via the *splitting matrix*

$$S(x_i) = -2\lambda_i \sum_{p=1}^{P} r_p \nabla_x^2 f_p(\boldsymbol{\theta}; x_i), \qquad (6)$$

where $r_p = \boldsymbol{\theta} - \sum_{i=1}^{k} \lambda_i f_p(\boldsymbol{\theta}; x_i)$ and $f_p(\cdot)$ is the $p$-th element of vector-valued function $f(\cdot)$. As shown in Appendix B.2, the minimum eigenvalue of $S(x_i)$ serves as a reliable proxy for directions of negative curvature of the full Hessian with respect to $x$.

If $\lambda_{\min}(S(x_i))$ is sufficiently negative, we split $x_i$ into two off-springs along the corresponding eigenvector:

$$x_i^{\pm} = x_i \pm \eta v_{\min}(S(x_i)), \quad \lambda_i^{\pm} = \tfrac{1}{2}\lambda_i,$$

where $\eta > 0$ is a small step size. In practice, we impose an upper bound $\eta \le \eta_{\max}$ and select $\eta$ via a line search to ensure a sufficient decrease in the reconstruction objective. A second-order Taylor expansion shows that this operation yields a strict decrease in the objective whenever $\lambda_{\min}(S(x_i)) < 0$, and the optimality of this splitting strategy is also proven (Appendix B.1). The stopping criteria for sample splitting step is set as $\lambda_{\min}(S(x_i)) > -\epsilon_H, \forall i$ for some stopping threshold $\epsilon_H > 0$.

The full procedure is summarized in Algorithm 1.

## 5.3. Convergence Analysis

We analyze the convergence of the proposed sample splitting algorithm. Our goal is to show that the algorithm converges to an approximate second-order stationary point with respect to the reconstructed samples $x := (x_1^{\top}, \ldots, x_k^{\top})^{\top} \in \mathbb{R}^{kd}$, while treating the coefficients $\lambda = (\lambda_1, \ldots, \lambda_k)^{\top}$ as auxiliary variables.

**Stationarity with respect to $x$.** Although optimization is performed jointly over $(x, \lambda)$, the coefficients $\lambda$ admit a closed-form least-squares minimizer given $x$. Since the reconstructed samples $x$ are the ultimate target, we focus on stationarity with respect to $x$.

**Assumption 5.1.** Recall the reconstruction objective (5). We assume that $L(x, \lambda)$ is three times differentiable in $x$ and satisfies $\|\nabla_x^2 L(x, \lambda)\| \le l$, $\|\nabla_x^3 L(x, \lambda)\| \le \rho$ for all $x \in \mathbb{R}^{kd}, \lambda \in \mathbb{R}^k$.

**Definition 5.2** (Approximate second-order stationarity)**.** Under Assumption 5.1, given $\epsilon > 0$, a point $(x, \lambda)$ is an $\epsilon$-second-order stationary point with respect to $x$ if

$$\|\nabla_x L(x, \lambda)\| \le \epsilon, \qquad \lambda_{\min}(\nabla_x^2 L(x, \lambda)) \ge -\sqrt{\rho\epsilon}.$$

This definition follows the classical notion of approximate second-order stationarity for Hessian-Lipschitz nonconvex objectives(Nesterov & Polyak, 2006).

---

**Algorithm 1** Sample Splitting Algorithm

---

**Input:** Initial samples $\{x_i\}_{i=1}^k$, weights $\{\lambda_i\}_{i=1}^k$, splitting threshold $\lambda_* < 0$, maximum splitting step size $\eta_{\max} > 0$.
**Output:** Reconstructed samples $\{x_i\}$ and weights $\{\lambda_i\}$.
**repeat**
  **Phase I: First-order descent.**
  Update $(x, \lambda)$ using gradient-based optimization
    until $\|\nabla_x L\| \le \epsilon$.
  **Phase II: Sample Splitting.**
  **for** $i = 1$ **to** $k$ **do**
    Compute splitting matrix $S(x_i)$.
    Compute $\lambda_{\min}(S(x_i))$.
  **end for**
  **if** $\min_i \lambda_{\min}(S(x_i)) < \lambda_*$ **then**
    Select $i$ such that $\lambda_{\min}(S(x_i)) < \lambda_*$.
    Let $v_{\min}(S(x_i))$ be the corresponding eigenvector.
    Choose $\eta \le \eta_{\max}$ by line search.
    **Sample splitting:**
    $x_i^+ \leftarrow x_i + \eta v_{\min}(S(x_i))$
    $x_i^- \leftarrow x_i - \eta v_{\min}(S(x_i))$
    $\lambda_i^+ \leftarrow \tfrac{1}{2}\lambda_i, \quad \lambda_i^- \leftarrow \tfrac{1}{2}\lambda_i$
  **end if**
**until** a stopping criterion is reached

---

**Surrogate curvature via splitting matrices.** Direct evaluation of the full Hessian $\nabla_x^2 L$ is computationally expensive. Instead, the algorithm relies on per-sample splitting matrices $S(x_i)$ which capture local curvature directions relevant to splitting. As shown in Appendix B.2, the minimum eigenvalue of the full Hessian satisfies

$$\lambda_{\min}(\nabla_x^2 L(x, \lambda)) \ge \min_{i=1,\ldots,k} \lambda_{\min}(S(x_i)). \qquad (7)$$

We now state the main convergence result of the proposed algorithm.

**Theorem 5.3.** *Under Assumption 5.1, suppose initial reconstruction loss is $L_0$. Let $\eta_g \le 1/l$ be the gradient descent step size, $\eta \le \frac{3}{2}\sqrt{\epsilon/\rho}$ the splitting step size, and $\epsilon_H = \sqrt{\rho\epsilon}$ be the stopping threshold. Then the proposed algorithm will reach an $\epsilon$-second-order stationary point in at most*

$$O\left(\frac{L_0}{\sqrt{\rho}\eta^2}\epsilon^{-1/2}\right)$$

*splitting steps and $O(\epsilon^{-2})$ total iterations.*

*Proof sketch.* The algorithm alternates between two phases.

*Phase I (first-order descent).* While $\|\nabla_x L(x, \lambda)\| > \epsilon$, standard gradient descent guarantees a sufficient decrease

$$L(x^{t+1}, \lambda^{t+1}) \le L(x^t, \lambda^t) - \frac{\eta_g}{2}\|\nabla_x L(x^t, \lambda^t)\|^2,$$

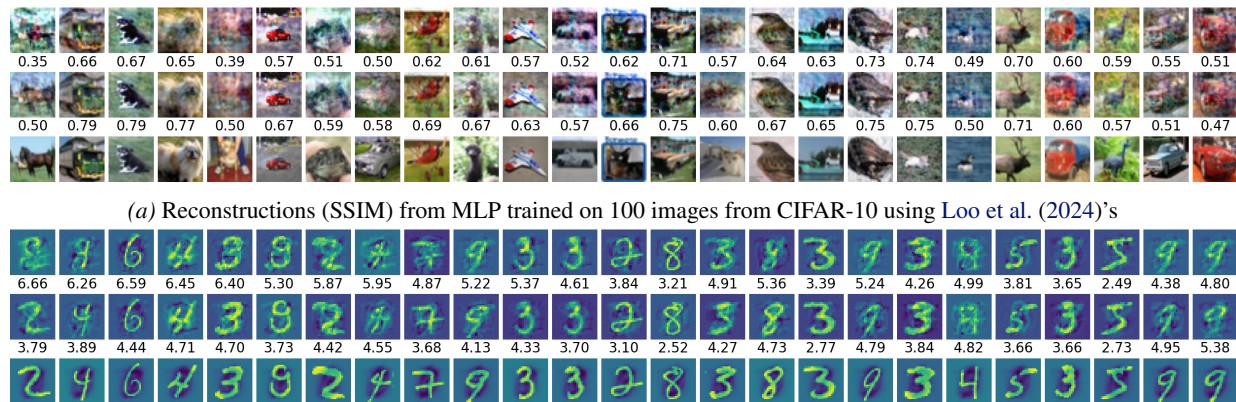

*(a)* Reconstructions (SSIM) from MLP trained on 100 images from CIFAR-10 using Loo et al. (2024)'s

*(b)* Reconstructions (L2 distance) from MLP trained on 100 images from MNIST using Loo et al. (2024)'s

*Figure 1.* Top 25 images reconstructed from MLP trained on 100 images using Loo et al. (2024)'s (row 1), Loo et al. (2024)'s with sample splitting (row 2) , and corresponding nearest neighbors from the dataset (row 3).

Since $L$ is bounded below, Phase I can only execute a finite number of iterations before the gradient norm becomes small.

*Phase II (sample splitting).* When $\|\nabla_x L(x, \lambda)\| \leq \epsilon$, the algorithm evaluates the splitting matrices. If $\lambda_{\min}(S(x_i)) \geq -\epsilon_H$ for all $i$, then by the surrogate-to-Hessian comparison (7), the point already satisfies approximate second-order stationarity.

Otherwise, splitting a sample with $\lambda_{\min}(S(x_i)) < -\epsilon_H$ yields a strict decrease

$$L_{\text{after split}} \leq L_{\text{before split}} - \frac{\eta^2}{2}\lambda_{\min}(S(x_i)),$$

where $\eta$ is the splitting step size. Since each split produces a quantifiable decrease and $L$ is bounded below, only finitely many splits can occur.

Combining the two phases, the algorithm cannot cycle indefinitely and must terminate at an approximate second-order stationary point. The detailed derivation is deferred to Appendix B.3.

**Remark.** Our convergence analysis can be related to existing results on non-convex optimization. It is well known that gradient descent guarantees $\|\nabla_x L\| \leq \epsilon$ within $O(\epsilon^{-2})$ iterations under standard smoothness assumptions. The proposed sample splitting strategy achieves an additional second-order-type guarantee, without requiring explicit computation of the full Hessian matrix, and within the same order of iterations. This is conceptually related to the results of Carmon & Duchi (2019), where approximate second-order stationarity is obtained via cubic regularization using Hessian-vector products. Moreover, our splitting criterion can detect directions of improvement even when the Hessian is positive semidefinite, highlighting that splitting stability captures complementary geometric information beyond standard Hessian-based analysis.

## 6. Experiments

In this section, we empirically evaluate the proposed sample splitting algorithm on image reconstruction tasks. Since splitting is a generic optimization technique, we demonstrate its effect using three representative reconstruction methods under different settings. Each subsection focuses on a specific aspect of the behavior of splitting, while additional results are deferred to the appendix.

### 6.1. Experiment setup

**Datasets.** We evaluate our methods on two standard image classification datasets: MNIST and CIFAR-10. For each dataset, we randomly sample a small subset of training images, which are treated as the unknown training set to be reconstructed. All images are normalized by subtracting the dataset mean.

**Models.** Following Haim et al. (2022), we consider a fully connected network with architecture $d$–1000–1000–1, where $d$ denotes the input dimension. Models are trained to convergence under either binary or multiclass classification settings with balanced labels, and the final parameters are used for reconstruction.

**Reconstruction Methods.** We consider three reconstruction methods: the KKT-based approaches of Haim et al. (2022) and Buzaglo et al. (2024), and the NTK-based method of Loo et al. (2024). Each method is evaluated with and without sample splitting under identical initialization and optimization settings. For simplicity, sample splitting is applied periodically after a fixed number of gradient descent iterations. Reconstruction quality is measured by SSIM on CIFAR-10 and L2 distance on MNIST.

## 6.2. Reconstruction performance

We first evaluate reconstruction quality using Loo et al. (2024)'s method on a binary classification task. Since this approach operates in the NTK regime and is most stable at relatively small scales, we reconstruct training sets of size 100 for both MNIST and CIFAR-10.

Figure 1 compares reconstructions with and without sample splitting. To focus on meaningful reconstructions, we select the top 25 training samples whose metric exceeds a fixed threshold either before or after splitting, and sort them by metric improvement. On CIFAR-10, 21 out of 25 samples achieve higher SSIM after splitting; on MNIST, 21 out of 25 samples achieve lower L2 distance. To assess the effect over the full training set, Figure 2 reports a per-training sample comparison. On CIFAR-10, improvements are concentrated among samples with higher baseline SSIM, while poorly reconstructed samples change little. On MNIST, the majority of samples show improvement.

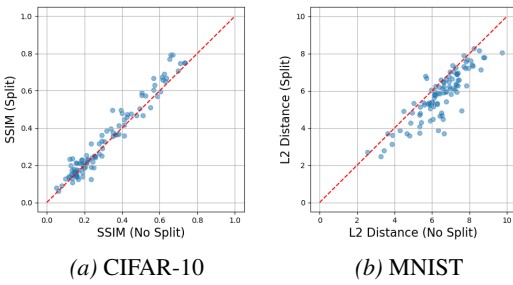

*(a)* CIFAR-10     *(b)* MNIST

*Figure 2.* Per-sample comparison of reconstruction metrics before and after splitting using Loo et al. (2024)'s method. Each point corresponds to a training sample, with the horizontal axis denoting the metric value without splitting and the vertical axis denoting the metric value with splitting.

## 6.3. Effect of splitting

**Optimization Dynamics.** We next study how splitting affects optimization dynamics using the method in Haim et al. (2022) on a binary classification task with 500 training samples. Figure 3 shows the evolution of reconstruction loss and evaluation metrics. Without splitting, the loss decreases steadily but slowly, while the metric improves early and then degrades, indicating drift away from the target data distribution. After splitting is applied, the metric decreases again near splitting events, improving reconstruction quality. This behavior is observed across different initial reconstruction sizes, suggesting that the effect of splitting is relatively robust.

**Per-Sample Trajectory Analysis** Finally, we visualize the effect of splitting at the level of individual samples using the multiclass reconstruction method of Buzaglo et al. (2024), with 500 training samples. Figure 4 tracks the op-

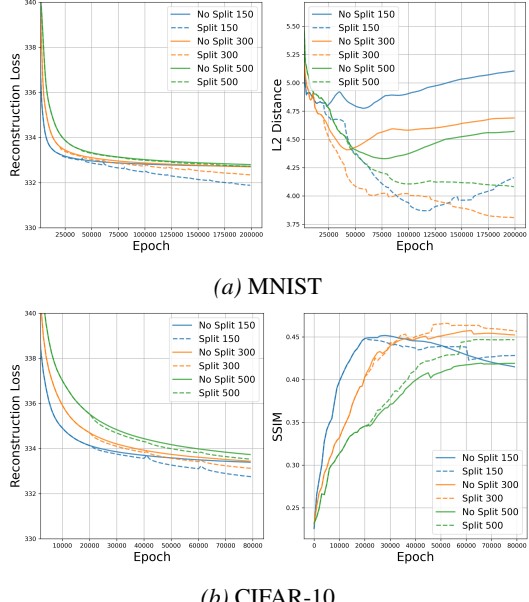

*(a)* MNIST

*(b)* CIFAR-10

*Figure 3.* Reconstruction loss and metrics for Haim et al. (2022)'s method with and without sample splitting using 500 training samples, with different initial reconstruction sizes per class.

timization trajectory of a representative reconstructed image. In the CIFAR-10 example, splitting is triggered at epoch 30,000, after which the SSIM exhibits a consistent increase. In contrast, without splitting, the SSIM continues to decrease. This example suggests that splitting can refine ambiguous reconstructions by generating nearby candidates, allowing further progress in cases where gradient-based updates alone appear to stagnate.

## 7. Discussion

This work studies data reconstruction from two complementary lens: identifiability and optimization. On the theoretical side, we show that for two-layer networks with polynomial activations of degree at least three, the KKT system uniquely determines the training samples under mild conditions, offering a principled explanation for the empirical success of prior reconstruction methods. On the algorithmic side, we introduce sample splitting as a lightweight refinement strategy that leverages second-order information to improve optimization in nonconvex reconstruction landscapes. Empirically, we demonstrate that sample splitting can enhance reconstruction quality across several existing methods.

As future work, we would like to extend the identifiability results to a robust analysis (approximate reconstruction) and partial reconstruction. We conjecture that when the interpolation condition is violated or even in the under-determined regime (network width $m$ smaller than the number of total unknowns), samples associated with spectrally isolated

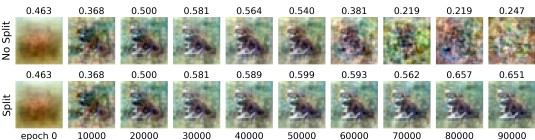

*(a)* CIFAR-10 (measured by SSIM; splitting checked every 30000 epochs)

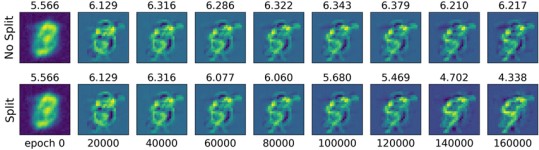

*(b)* MNIST (measured by L2 distance; splitting checked every 40000 epochs)

*Figure 4.* Optimization trajectories of a representative reconstructed sample using Buzaglo et al. (2024)'s method, with and without sample splitting.

eigenmodes of the induced tensor can be approximately reconstructed even when full recovery is impossible.

## Acknowledgements

This work is supported by NSF DMS-2523382.

## Impact Statement

We study training-data reconstruction from an attacker's viewpoint, from which one can better quantify privacy leakage. In security and privacy, meaningful mitigation requires adversarial threat models and worst-case evaluation—weak attacks can understate risk. We provide identifiability conditions for KKT-based reconstruction and a general optimization refinement that strengthens reconstruction across objectives, enabling more faithful auditing of privacy exposure and potentially informing stronger defenses.

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

# A. Proofs for Identifiability

## A.1. Preliminaries and notation

Throughout this section, we fix $\alpha \geq 3$ and an active set $S := \{i \in [n] : \lambda_i > 0\}$. Recall the signed weights $b_i := \lambda_i y_i$ and the symmetric order-$\alpha$ tensor

$$\mathcal{T} := \sum_{i \in S} b_i \, x_i^{\otimes \alpha}. \tag{8}$$

Define the degree-$(\alpha - 1)$ vector-valued homogeneous polynomial map (a tensor contraction)

$$f(w) := \mathcal{T}(\cdot, w, \ldots, w) = \sum_{i \in S} b_i (x_i^\top w)^{\alpha - 1} x_i \qquad (w \in \mathbb{R}^d). \tag{9}$$

Let $N := \binom{d + \alpha - 2}{\alpha - 1}$ be the dimension of the space of homogeneous polynomials of degree $\alpha - 1$ in $d$ variables.

We also use the homogeneous polynomial kernel of degree $\alpha - 1$:

$$\kappa(u, v) := (u^\top v)^{\alpha - 1}. \tag{10}$$

Given $W_1, \ldots, W_m$, define the Gram matrix $K \in \mathbb{R}^{m \times m}$ by

$$K_{pq} := \kappa(W_p, W_q) = (W_p^\top W_q)^{\alpha - 1}. \tag{11}$$

For any query $w \in \mathbb{R}^d$, define the kernel vector $k(w) \in \mathbb{R}^m$ by

$$k(w) := \big(\kappa(W_1, w), \ldots, \kappa(W_m, w)\big)^\top. \tag{12}$$

## A.2. Proof of Theorem 4.1

With the notations introduced above, we give a detailed proof of Theorem 4.1

*Proof of Theorem 4.1.* We prove the two claims in two steps.

**Step 1:** $(a, W)$ **uniquely determines** $f$ **and** $\mathcal{T}$. By the KKT stationarity equations eq. (1), for every neuron $j \in [m]$ we have the point-evaluation identity

$$W_j = \alpha \, a_j \, f(W_j). \tag{13}$$

Hence, whenever $a_j \neq 0$, the value $f(W_j)$ is directly known from $(a, W)$:

$$f(W_j) = \frac{1}{\alpha a_j} W_j. \tag{14}$$

(If $a_j = 0$, then (13) forces $W_j = 0$ and thus $f(W_j) = f(0) = 0$ is also known.)

*Uniqueness of interpolation.* Let $\phi : \mathbb{R}^d \to \mathbb{R}^N$ be the standard monomial feature map for homogeneous polynomials of degree $\alpha - 1$. Since each coordinate of $f$ is such a polynomial, there exists a coefficient matrix $A \in \mathbb{R}^{d \times N}$ such that

$$f(w) = A \, \phi(w). \tag{15}$$

Define the feature matrix $V \in \mathbb{R}^{m \times N}$ by $V_{j,:}^\top = \phi(W_j)$ and the value matrix $F \in \mathbb{R}^{m \times d}$ by $F_{j,:}^\top = f(W_j)$. Then (15) yields

$$F = V A^\top. \tag{16}$$

Moreover, by construction of the polynomial kernel, we have

$$K = V V^\top. \tag{17}$$

Assume the interpolation condition (2), i.e. $\mathrm{rank}(K) = N$. Since $K = V V^\top$ and $\mathrm{rank}(K) = \mathrm{rank}(V)$, we have $\mathrm{rank}(V) = N$. In particular, $V$ has full column rank and (16) has a unique solution $A$.

Equivalently, one may express the unique interpolant in kernel form. Since $V$ has full column rank, its Moore–Penrose pseudoinverse is

$$V^\dagger = V^\top (VV^\top)^\dagger = V^\top K^\dagger. \tag{18}$$

Thus from (16) we get

$$A^\top = V^\dagger F = V^\top K^\dagger F, \qquad \text{so} \qquad f(w) = A\phi(w) = F^\top K^\dagger k(w). \tag{19}$$

This shows $f$ is uniquely determined by the evaluations (14), hence by $(a, W)$.

*f uniquely determines $\mathcal{T}$.* Define the scalar homogeneous polynomial of degree $\alpha$:

$$p(w) := \langle w, f(w) \rangle. \tag{20}$$

Using (9), we have

$$p(w) = \sum_{i \in S} b_i (x_i^\top w)^{\alpha-1} \langle w, x_i \rangle = \sum_{i \in S} b_i (x_i^\top w)^\alpha = \mathcal{T}(w, \dots, w). \tag{21}$$

Hence knowledge of $f$ implies knowledge of $p(w) = \mathcal{T}(w, \dots, w)$ for all $w$.

Finally, the symmetric multilinear form $\mathcal{T}$ is uniquely determined by its diagonal polynomial $p(w) = \mathcal{T}(w, \dots, w)$. By calculation, for any $u_1, \dots, u_\alpha \in \mathbb{R}^d$,

$$\mathcal{T}(u_1, \dots, u_\alpha) = \frac{1}{\alpha!\, 2^\alpha} \sum_{\varepsilon \in \{\pm 1\}^\alpha} \Big( \prod_{t=1}^\alpha \varepsilon_t \Big)\, p\Big( \sum_{t=1}^\alpha \varepsilon_t u_t \Big). \tag{22}$$

Therefore, $\mathcal{T}$ is uniquely determined by $f$, and thus uniquely determined by $(a, W)$.

**Step 2: recovering $\{(x_i, b_i)\}_{i \in S}$ from $\mathcal{T}$.** Let $r := |S|$ and recall

$$\mathcal{T} = \sum_{i \in S} b_i x_i^{\otimes \alpha}, \qquad f(w) = \sum_{i \in S} b_i (x_i^\top w)^{\alpha-1} x_i. \tag{23}$$

Define the symmetric matrix slice

$$M(v) := \mathcal{T}(\cdot, \cdot, v, \dots, v) = \sum_{i \in S} \gamma_i(v)\, x_i x_i^\top, \qquad \gamma_i(v) := b_i (x_i^\top v)^{\alpha-2}. \tag{24}$$

Note $M(v)$ can be indefinite when the $\gamma_i(v)$ have mixed signs. The argument here does not require $M(v) \succeq 0$.

Assume the active directions $\{x_i\}_{i \in S}$ are linearly independent, so $r \le d$ and $X := [x_i]_{i \in S} \in \mathbb{R}^{d \times r}$ has full column rank. Draw $v_1, v_2 \in \mathbb{R}^d$ randomly from any absolutely continuous distribution (e.g. uniform distribution), Then

$$x_i^\top v_1 \ne 0, \quad x_i^\top v_2 \ne 0 \quad (\forall i \in S), \qquad \frac{(x_i^\top v_2)^{\alpha-2}}{(x_i^\top v_1)^{\alpha-2}} \ne \frac{(x_{i'}^\top v_2)^{\alpha-2}}{(x_{i'}^\top v_1)^{\alpha-2}} \quad (\forall i \ne i'). \tag{25}$$

holds almost surely.

Compute

$$A := M(v_1), \qquad B := M(v_2), \tag{26}$$

which are computable from $\mathcal{T}$ via (24). Let $U \in \mathbb{R}^{d \times r}$ be any orthonormal basis of $\mathrm{range}(A) = \mathrm{span}\{x_i\}_{i \in S}$ (e.g. from an SVD of $A$). Define

$$\bar{A} := U^\top A U, \qquad \bar{B} := U^\top B U \in \mathbb{R}^{r \times r}. \tag{27}$$

Since $X$ has full column rank and $\gamma_i(v_1) \ne 0$, we have $\mathrm{rank}(A) = r$, hence $\bar{A}$ is invertible.

Writing $D_\ell := \mathrm{diag}(\gamma_i(v_\ell))_{i \in S}$ for $\ell \in \{1, 2\}$, we have

$$A = XD_1 X^\top, \qquad B = XD_2 X^\top. \tag{28}$$

Let $G := U^\top X \in \mathbb{R}^{r \times r}$; since $U$ spans $\mathrm{range}(X)$, $G$ is invertible and

$$\bar{A} = GD_1 G^\top, \qquad \bar{B} = GD_2 G^\top. \tag{29}$$

Consider

$$C := \bar{A}^{-1} \bar{B}. \tag{30}$$

Using (29),

$$C = (G^\top)^{-1} D_1^{-1} D_2 \, G^\top, \tag{31}$$

so $C$ is similar to the diagonal matrix $D := D_1^{-1} D_2$ with entries

$$D_{ii} = \frac{\gamma_i(v_2)}{\gamma_i(v_1)} = \frac{(x_i^\top v_2)^{\alpha-2}}{(x_i^\top v_1)^{\alpha-2}}. \tag{32}$$

By (25), these are pairwise distinct, hence $C$ has $r$ distinct real eigenvalues and is diagonalizable. Let $C = V\Lambda V^{-1}$ with $V$ invertible. Comparing with (31), we get that $V^{-\top}$ equals $G$ up to permutation and scaling. Therefore, defining

$$\widehat{X} := UV^{-\top} \in \mathbb{R}^{d \times r}, \tag{33}$$

we obtain

$$\widehat{X} = X\Pi\Delta \tag{34}$$

for some permutation matrix $\Pi$ and some invertible diagonal matrix $\Delta$. Thus the columns of $\widehat{X}$ recover the active directions $\{x_i\}_{i \in S}$ up to permutation and scaling.

Since the model fixes a scale for $x_i$, normalize each recovered column:

$$\tilde{x}_k := \hat{x}_k / \|\hat{x}_k\|_2. \tag{35}$$

Then contract $\mathcal{T}$ to recover the coefficient:

$$\tilde{b}_k := \mathcal{T}(\tilde{x}_k, \ldots, \tilde{x}_k) = \sum_{i \in S} b_i (x_i^\top \tilde{x}_k)^\alpha = b_{\pi(k)}. \tag{36}$$

Combining Step 1 and Step 2 completes the proof: $(a, W)$ uniquely determines $\mathcal{T}$, and from $\mathcal{T}$ we recover $\{(x_i, b_i)\}_{i \in S}$ up to permutation almost surely. $\qquad \square$

### A.3. Checking the assumptions for non-homogeneous activations

**Notations.** In this subsection we denote the network output by $\Phi_\sigma(\theta; x)$, where $\theta = (a, W) \in \mathbb{R}^m \times \mathbb{R}^{m \times d}$ and

$$\Phi_\sigma(\theta; x) = \sum_{j=1}^m a_j \, \sigma(w_j^\top x), \qquad \sigma(t) = \sum_{k=0}^\alpha c_k t^k, \quad c_\alpha \neq 0, \ \alpha \geq 3. \tag{37}$$

Assume the data are bounded: $\|x_i\|_2 \leq R$.

We train with an exponential-type loss (as in Cai et al. (2025)):

$$L(\theta) = \frac{1}{n} \sum_{i=1}^n \exp\left( - y_i \, \Phi_\sigma(\theta; x_i) \right). \tag{38}$$

**Homogenization equals the top-degree part.** Under uniform scaling $\theta \mapsto r\theta$, the leading homogeneous component is

$$\Phi_\mathrm{H}(\theta; x) := \lim_{r \to \infty} \frac{\Phi_\sigma(r\theta; x)}{r^M} \quad \text{with} \quad M = \alpha + 1, \tag{39}$$

and for polynomial $\sigma$ this limit exists and equals the network with activation $\sigma_{\mathrm{H}}(t) = c_\alpha t^\alpha$:

$$\Phi_{\mathrm{H}}(\theta; x) = \sum_{j=1}^{m} a_j \, c_\alpha (w_j^\top x)^\alpha. \tag{40}$$

This is exactly the homogeneous model used in Section 4.2.

According to Cai et al. (2025), Theorem 3.5 implies that the limiting direction of gradient flow satisfies the KKT conditions of the max-margin problem defined by the homogenization $\Phi_{\mathrm{H}}$. To apply this theorem, the network should satisfy Assumptions 1-3 in Cai et al. (2025). Now we check the assumptions.

**Assumption 1: near-$M$-homogeneity.** Cai et al. (2025) require (Definition 1 / Assumption 1) that $f(\theta; x)$ is near-$M$-homogeneous, i.e.there exist polynomials $p, q$ of degree at most $M$ such that, for (sub)gradients $h \in \partial_\theta f(\theta; x)$,

$$\big|\langle h, \theta \rangle - M f(\theta; x)\big| \leq p'(\|\theta\|), \qquad \|h\| \leq q'(\|\theta\|), \qquad |f(\theta; x)| \leq q(\|\theta\|),$$

and the derived function $p_a$ satisfies $p_a(x)/x^{M-1} \to 0$.

In our setting, $\Phi_\sigma(\theta; x)$ is a polynomial in $\theta$ and can be decomposed as a sum of homogeneous parts:

$$\Phi_\sigma(\theta; x) = \sum_{k=0}^{\alpha} c_k \, \Phi^{(k)}(\theta; x), \qquad \Phi^{(k)}(\theta; x) := \sum_{j=1}^{m} a_j (w_j^\top x)^k, \tag{41}$$

where $\Phi^{(k)}$ is homogeneous of degree $(k+1)$ in $\theta$ under uniform scaling. By the property of homogeneous functions, $\langle \nabla_\theta \Phi^{(k)}(\theta; x), \theta \rangle = (k+1)\Phi^{(k)}(\theta; x)$. With $M = \alpha + 1$, this yields

$$\langle \nabla_\theta \Phi_\sigma(\theta; x), \theta \rangle - M \Phi_\sigma(\theta; x) = -\sum_{k=0}^{\alpha-1} (\alpha - k) \, c_k \, \Phi^{(k)}(\theta; x). \tag{42}$$

Since $\|x\| \leq R$, each $\Phi^{(k)}(\theta; x)$ admits a polynomial growth bound in $\|\theta\|$ of degree $(k+1) \leq \alpha$. Consequently, there exists a constant $C = C(\alpha, \{c_k\}, R, m)$ such that for all $(\theta, x)$,

$$\big|\langle \nabla_\theta \Phi_\sigma(\theta; x), \theta \rangle - (\alpha + 1)\Phi_\sigma(\theta; x)\big| \leq C \, (1 + \|\theta\|^\alpha). \tag{43}$$

Moreover, $\nabla_\theta \Phi_\sigma(\theta; x)$ is also a polynomial in $\theta$ of degree at most $\alpha$, hence

$$\|\nabla_\theta \Phi_\sigma(\theta; x)\| \leq C \, (1 + \|\theta\|^\alpha), \qquad |\Phi_\sigma(\theta; x)| \leq C \, (1 + \|\theta\|^{\alpha+1}). \tag{44}$$

Therefore, Assumption 1 of Cai et al. (2025) holds with $M = \alpha + 1$ by taking, e.g.,

$$p(t) = C \, (t + t^{\alpha+1}), \qquad q(t) = C \, (1 + t + t^{\alpha+1}). \tag{45}$$

**Assumption 3: weak-homogeneous gradient.** Assumption 3 requires that the gradient asymptotically matches that of the homogenization:

$$\lim_{r \to \infty} \left\| \frac{\nabla_\theta \Phi_\sigma(r\theta; x)}{r^{M-1}} - \nabla_\theta \Phi_{\mathrm{H}}(\theta; x) \right\| = 0 \qquad (M = \alpha + 1), \tag{46}$$

uniformly in the sense specified in Cai et al. (2025). For polynomial $\Phi_\sigma$, (46) follows by direct degree counting: $\nabla_\theta \Phi_\sigma(r\theta; x)$ is a polynomial in $r$ of maximum degree $(M - 1) = \alpha$, whose leading coefficient equals $\nabla_\theta \Phi_{\mathrm{H}}(\theta; x)$, and all lower-degree terms vanish after division by $r^\alpha$.

**Assumption 2: strong separability.** Assumption 2 requires that there exists some $s > 0$ such that

$$L(\theta_s) < \frac{1}{n} \exp\big(-p_a(\|\theta_s\|)\big), \tag{47}$$

where $p_a$ is the function induced by $p$ in Assumption 1. Using (38), note that

$$\min_{i \in [n]} y_i \Phi_\sigma(\theta; x_i) \geq \log \frac{1}{nL(\theta)}. \tag{48}$$

Thus (47) is equivalent to requiring $\log \frac{1}{nL(\theta_s)} > p_a(\|\theta_s\|)$. In particular, Assumption 2 is ensured when the homogenized model $\Phi_{\mathrm{H}}$ can separate the data with positive margin or the optimization dynamics indeed drives $L(\theta_t)$ sufficiently small.

# B. Proofs for the Sample Splitting Algorithm

## B.1. Taylor Expansion for Sample Splitting and Optimal Splitting Strategy

**Lemma B.1** (Infinitesimal Splitting Expansion). *Under Assumption 5.1, consider* $x_i^{[j]} = x_i + \eta \delta_i^{[j]}$, $\sum_{j=1}^{k_i} \lambda_i^{[j]} \delta_i^{[j]} = 0, \forall i = 1, \cdots, k$, *where* $\|\delta_i^{[j]}\| \leq 1$ *denote infinitesimal off-spring displacements. Define the reconstruction residual* $r = \boldsymbol{\theta} - \sum_{i=1}^{k} \lambda_i f(\boldsymbol{\theta}; x_i)$, *and the splitting matrix for sample* $x_i$ *as*

$$S(x_i) = -2\lambda_i \sum_{p=1}^{P} r_p \nabla_{x_i}^2 f_p(\boldsymbol{\theta}; x_i).$$

*Then the reconstructed loss after infinitesimal splitting admits the expansion*

$$L(\{\boldsymbol{x}_i\}_{i=1}^k, \{\boldsymbol{\lambda}_i\}_{i=1}^k) \leq L(\{x_i\}_{i=1}^k, \{\lambda_i\}_{i=1}^k) + \frac{\eta^2}{2} \sum_{i=1}^k \sum_{j=1}^{k_i} \frac{\lambda_i^{[j]}}{\lambda_i} \delta_i^{[j]\top} S(x_i) \delta_i^{[j]} + \frac{\rho\eta^3}{6}$$

*Here, the second term captures the contribution of the splitting directions* $\delta$ *and depends on* $x_i$ *only through its corresponding splitting matrix* $S(x_i)$.

*Proof.* Denote $J(x_i) := \nabla_{x_i} f(\boldsymbol{\theta}; x_i)$, $T(x_i) := T(x_i, x_i) = 2\lambda_i^2 J(x_i)^\top J(x_i)$ and $T(x_i, x_j) := 2\lambda_i \lambda_j J(x_i)^\top J(x_j), i \neq j$. By direct calculation, the gradient and Hessian of reconstruction loss $L(\{x_i\}_{i=1}^k, \{\lambda_i\}_{i=1}^k) = \|\boldsymbol{\theta} - \sum_{i=1}^k \lambda_i f(\boldsymbol{\theta}; x_i)\|^2$ satisfy

$$\nabla_{x_i} L(\{x_i\}_{i=1}^k, \{\lambda_i\}_{i=1}^k) = -2\lambda_i \nabla_{x_i} f(\boldsymbol{\theta}; x_i)^\top r = -2J(x_i)^\top r,$$

$$\nabla_{x_i}^2 L(\{x_i\}_{i=1}^k, \{\lambda_i\}_{i=1}^k) = 2\lambda_i^2 \nabla_{x_i} f(\boldsymbol{\theta}; x_i)^\top \nabla_{x_i} f(\boldsymbol{\theta}; x_i) - 2\lambda_i \sum_{p=1}^{P} r_p \nabla_{x_i}^2 f_p(\boldsymbol{\theta}; x_i) = T(x_i) + S(x_i).$$

After splitting, the augmented loss is $L(\{\boldsymbol{x}_i\}_{i=1}^k, \{\boldsymbol{\lambda}_i\}_{i=1}^k) = \|\boldsymbol{\theta} - \sum_{i=1}^k \sum_{j=1}^{k_i} \lambda_i^{[j]} f(\boldsymbol{\theta}; x_i^{[j]})\|^2$. As the weights satisfy $\sum_{j=1}^{k_i} \lambda_i^{[j]}/\lambda_i = 1, \lambda_i^{[j]} > 0$, we have $L(\{\mathbf{1}_{m_i} \otimes x_i\}_{i=1}^k, \{\boldsymbol{\lambda}_i\}_{i=1}^k) = L(\{x_i\}_{i=1}^k, \{\lambda_i\}_{i=1}^k)$. Taking the gradient of $L(\{\boldsymbol{x}_i\}_{i=1}^k, \{\boldsymbol{\lambda}_i\}_{i=1}^k)$ when $\boldsymbol{x}_i = \mathbf{1}_{k_i} \otimes x_i$ i.e. $x_i^{[j]} = x_i, j = 1, \ldots, k_i$, we have

$$\nabla_{x_i^{[j]}} L(\{\boldsymbol{x}_i\}_{i=1}^k, \{\boldsymbol{\lambda}_i\}_{i=1}^k) = -2\lambda_i^{[j]} \nabla_{x_i^{[j]}} f(\boldsymbol{\theta}; x_i^{[j]})^\top r = \frac{\lambda_i^{[j]}}{\lambda_i} \nabla_{x_i} L(\{x_i\}_{i=1}^k, \{\lambda_i\}_{i=1}^k),$$

$$\nabla_{x_i^{[j]}}^2 L(\{\boldsymbol{x}_i\}_{i=1}^k, \{\boldsymbol{\lambda}_i\}_{i=1}^k) = 2(\lambda_i^{[j]})^2 \nabla_{x_i^{[j]}} f(\boldsymbol{\theta}; x_i^{[j]})^\top \nabla_{x_i^{[j]}} f(\boldsymbol{\theta}; x_i^{[j]}) - 2\lambda_i^{[j]} \sum_{p=1}^{P} r_p \nabla_{x_i^{[j]}}^2 f_p(\boldsymbol{\theta}; x_i^{[j]}) = (\frac{\lambda_i^{[j]}}{\lambda_i})^2 T(x_i) + \frac{\lambda_i^{[j]}}{\lambda_i} S(x_i).$$

For $i \neq r$ or $j \neq s$,

$$\nabla_{x_i^{[j]}, x_r^{[s]}} L(\{\boldsymbol{x}_i\}_{i=1}^k, \{\boldsymbol{\lambda}_i\}_{i=1}^k) = 2(\lambda_i^{[j]} \lambda_r^{[s]}) \nabla_{x_i^{[j]}} f(\boldsymbol{\theta}; x_i^{[j]})^\top \nabla_{x_r^{[s]}} f(\boldsymbol{\theta}; x_r^{[s]}) = \frac{\lambda_i^{[j]} \lambda_r^{[s]}}{\lambda_i \lambda_r} T(x_i, x_r).$$

Note that $x_i^{[j]} = x_i + \eta\delta_i^{[j]}$, $\sum_{j=1}^{k_i} \lambda_i^{[j]}\delta_i^{[j]} = 0$, $\sum_{j=1}^{k_i} \lambda_i^{[j]} = \lambda_i, \forall i = 1, \cdots, m$, we obtain the Taylor expansion at $\eta = 0$:

$$L(\{\boldsymbol{x}_i\}_{i=1}^k, \{\boldsymbol{\lambda}_i\}_{i=1}^k) - L(\{x_i\}_{i=1}^k, \{\lambda_i\}_{i=1}^k) = L(\{\mathbf{1}_{k_i} \otimes x_i + \eta\boldsymbol{\delta}_i\}_{i=1}^k, \{\boldsymbol{\lambda}_i\}_{i=1}^k) - L(\{x_i\}_{i=1}^k, \{\lambda_i\}_{i=1}^k)$$

$$\leq \eta \sum_{i=1}^k \sum_{j=1}^{k_i} \nabla_{x_i^{[j]}} L(\{\mathbf{1}_{k_i} \otimes x_i\}_{i=1}^k, \{\boldsymbol{\lambda}_i\}_{i=1}^k)^\top \delta_i^{[j]}$$

$$+ \frac{\eta^2}{2} \sum_{i,r=1}^k \sum_{j=1}^{k_i} \sum_{s=1}^{k_r} \delta_i^{[j]\top} \nabla_{x_i^{[j]}, x_r^{[s]}} L(\{\mathbf{1}_{k_i} \otimes x_i\}_{i=1}^k, \{\boldsymbol{\lambda}_i\}_{i=1}^k) \delta_r^{[s]} + \frac{\rho\eta^3}{6}$$

$$= \eta \sum_{i=1}^k \sum_{j=1}^{k_i} \frac{\lambda_i^{[j]}}{\lambda_i} \nabla_{x_i} L(\{x_i\}_{i=1}^k, \{\lambda_i\}_{i=1}^k)^\top \delta_i^{[j]} + \frac{\eta^2}{2} \sum_{i=1}^k \sum_{j=1}^{k_i} \frac{\lambda_i^{[j]}}{\lambda_i} \delta_i^{[j]\top} S(x_i)\delta_i^{[j]}$$

$$+ \frac{\eta^2}{2} \sum_{i,r=1}^k \sum_{j=1}^{k_i} \sum_{s=1}^{k_r} \frac{\lambda_i^{[j]}\lambda_r^{[s]}}{\lambda_i\lambda_k} \delta_i^{[j]\top} \nabla_{x_i^{[j]}, x_r^{[s]}} T(x_i, x_r)\delta_r^{[s]} + \frac{\rho\eta^3}{6}$$

$$= \frac{\eta^2}{2} \sum_{i=1}^k \sum_{j=1}^{k_i} \frac{\lambda_i^{[j]}}{\lambda_i} (\delta_i^{[j]})^\top S(x_i)(\delta_i^{[j]}) + \frac{\rho\eta^3}{6},$$

where the last equation is because $\sum_{j=1}^{k_i} \lambda_i^{[j]}\delta_i^{[j]} = 0, \sum_{s=1}^{k_r} \lambda_r^{[s]}\delta_r^{[s]} = 0$.

We have

$$L(\{\boldsymbol{x}_i\}_{i=1}^k, \{\boldsymbol{\lambda}_i\}_{i=1}^k) \leq L(\{x_i\}_{i=1}^k, \{\lambda_i\}_{i=1}^k) + \frac{\eta^2}{2} \sum_{i=1}^k \sum_{j=1}^{k_i} \frac{\lambda_i^{[j]}}{\lambda_i} \delta_i^{[j]\top} S(x_i)\delta_i^{[j]} + \frac{\rho\eta^3}{6}$$

□

**Theorem B.2** (Optimal Infinitesimal Splitting Strategy). *Let $\lambda_{min}(S(x_i))$ denote the smallest eigenvalue of $S(x_i)$ and $v_{min}(x_i)$ its corresponding eigenvector. Then the optimal infinitesimal splitting strategy for each sample $x_i$ is given as follows.*

1. *Splitting-stable case: If $\lambda_{min}(S(x_i)) \geq 0$, then any infinitesimal splitting of $x_i$ cannot decrease the reconstruction loss.*

2. *Optimal splitting case: If $\lambda_{min}(S(x_i)) < 0$, the maximal decrease in loss subject to $\|\delta_i^{[j]}\| \leq 1$ is achieved by a binary split*

$$m_i = 2, \quad \lambda_i^{[1]} = \lambda_i^{[2]} = \frac{1}{2}\lambda_i, \quad and \quad \delta_i^{[1]} = v_{\min}(S(x_i)), \quad \delta_i^{[2]} = -v_{\min}(S(x_i)).$$

*In this case, the decrease in loss for splitting sample $x_i$ is $\frac{\eta^2}{2}\lambda_{\min}(S(x_i)) < 0$.*

*Proof.* By Lemma B.1, the second-order decrease in loss due to splitting sample $x_i$ is $\frac{\eta^2}{2} \sum_{j=1}^{k_i} \frac{\lambda_i^{[j]}}{\lambda_i} \delta_i^{[j]\top} S(x_i)\delta_i^{[j]}$ with $\|\delta_i^{[j]}\| = 1$. Since $\delta_i^{[j]\top} S(x_i)\delta_i^{[j]} \geq \lambda_{\min}(S(x_i))$ for all unit vectors $\delta_i^{[j]}$, we have

$$\frac{\eta^2}{2} \sum_{j=1}^{k_i} \frac{\lambda_i^{[j]}}{\lambda_i} \delta_i^{[j]\top} S(x_i)\delta_i^{[j]} \geq \frac{\eta^2}{2}\lambda_{\min}(S(x_i))$$

Equality is achieved when $k_i = 2, \lambda_i^{[1]} = \lambda_i^{[2]} = \frac{1}{2}\lambda_i, \delta_i^{[1]} = v_{\min}(S(x_i)), \delta_i^{[2]} = -v_{\min}(S(x_i))$. Note that additivity of the second-order loss decrease across samples implies that optimizing each sample independently yields a globally optimal splitting strategy. □

## B.2. Splitting Matrix and Negative Curvature

We relate the splitting matrix $S(x_i)$ to the curvature of the full Hessian $\nabla_x^2 L$.

**Lemma B.3.** *Let $H = \nabla_x^2 L(x, \lambda)$ be the Hessian with respect to $x$. Then for any $i$,*

$$\lambda_{\min}(H) \geq \min_i \lambda_{\min}(S(x_i)).$$

*where*

$$S(x_i) = -2\lambda_i \sum_{p=1}^{P} r_p \nabla_{x_i}^2 f_p(\boldsymbol{\theta}; x_i), \quad r_p = \boldsymbol{\theta} - \sum_{i=1}^{m} \lambda_i f_p(\boldsymbol{\theta}; x_i),$$

*and $f_p(\cdot)$ is the $p$-th element of vetor-valued function $f(\cdot)$.*

*Proof.* Denote $J_i := \nabla_{x_i} \lambda_i f(\boldsymbol{\theta}; x_i), H_{ij} := \nabla_{x_i, x_j} L(x, \lambda)$, then we have

$$H_{ii} = 2J_i^\top J_i + S(x_i), \quad H_{ij} = 2J_i^\top J_j, \ i \neq j.$$

Therefore, for any $v = (v_1, \ldots, v_k)$,

$$v^\top H v = 2\| \sum_{i=1}^{k} J_i v_i \|^2 + \sum_{i=1}^{k} v_i^\top S(x_i) v_i \geq \sum_{i=1}^{k} v_i^\top S(x_i) v_i$$

Therefore,

$$\lambda_{\min}(H) \geq \min_i \lambda_{\min}(S(x_i)).$$

$\square$

## B.3. Convergence Analysis

**Theorem B.4.** *Under Assumption 5.1, suppose initial reconstruction loss is $L_0$. Let $\eta_g \leq 1/l$ be the gradient descent step size, $\eta \leq \frac{3}{2}\sqrt{\epsilon/\rho}$ the splitting step size, and $\epsilon_H = \sqrt{\rho\epsilon}$ be the stopping threshold. Then the proposed algorithm will reach an $\epsilon$-second-order stationary point in at most*

$$O(\frac{L_0}{\sqrt{\rho}\eta^2}\epsilon^{-1/2})$$

*splitting steps and $O(\epsilon^{-2})$ total iterations.*

*Proof.* The algorithm alternates between two phases.

**Phase I (first-order descent).** While $\|\nabla_x L(x, \lambda)\| > \epsilon$, gradient descent with $\eta_g \leq 1/l$ ensures

$$
\begin{aligned}
L(x^{t+1}, \lambda^{t+1}) &\leq L(x^{t+1}, \lambda^t) \\
&\leq L(x^t, \lambda^t) + \nabla_x L(x^t, \lambda^t)^\top (x^{t+1} - x^t) + \frac{l}{2}\|x^{t+1} - x^t\|^2 \\
&= L(x^t, \lambda^t) - \eta_g\|\nabla_x L(x^t, \lambda^t)\|^2 + \frac{l\eta_g^2}{2}\|\nabla_x L(x^t, \lambda^t)\|^2 \\
&\leq L(x^t, \lambda^t) - \frac{\eta_g}{2}\|\nabla_x L(x^t, \lambda^t)\|^2,
\end{aligned}
$$

Since $L \geq 0$, Phase I can be executed at most $O(\frac{L_0}{\eta_g}\epsilon^{-2})$ iterations.

**Phase II (sample splitting).** When $\|\nabla_x L(x, \lambda)\| \leq \epsilon$, the algorithm evaluates the splitting matrices. If $\lambda_{\min}(S(x_i)) \geq -\epsilon_H$ for all $i$, the algorithm terminates.

Otherwise, there at least exist a sample $x_i$ with $\lambda_{\min}(S(x_i)) < -\epsilon_H$. Suppose that $\boldsymbol{x}^{[i]}$ is the sample vector we achieve by splitting only sample $x_i$, and $\boldsymbol{x}$ is the sample vector we achieve after the splitting step. If we adopt the splitting strategy in 1,

we have by Lemma B.3 and Theorem B.2 that

$$L(\boldsymbol{x}, \boldsymbol{\lambda}) - L(x, \lambda) \leq L(\boldsymbol{x}^{[i]}, \boldsymbol{\lambda}^{[i]}) - L(x, \lambda)$$

$$\leq \frac{\eta^2}{2}\lambda_{\min}(S(x_i)) + \frac{\rho}{6}\eta^3$$

$$\leq -\frac{\eta^2\epsilon_H}{2} + \frac{\rho}{6}\eta^3 \leq -\frac{\epsilon_H\eta^2}{4}$$

where the last inequality follows from the choice of $\eta$. Since $L \geq 0$, the algorithm must terminate within the following number of iterations of splitting:

$$\frac{L_0 - 0}{\epsilon_H\eta^2/4} = O(\frac{L_0}{\sqrt{\rho}\eta^2}\epsilon^{-1/2})$$

The total number of iterations is the summation of number of iterations in Phase I and Phase II, i.e.

$$2\frac{L_0}{\eta_g\epsilon^2} + \frac{L_0}{\epsilon_H\eta^2/4} = O(\epsilon^{-2}),$$

and does not depend explicitly on intrinsic dimension $d$.

Finally, we will ensure that when algorithm terminates, the point is $\epsilon$-second-order stationary. Upon termination, we have $\|\nabla_x L(x, \lambda)\| \leq \epsilon$ and $\lambda_{\min}(S(x_i)) \geq -\epsilon_H$ for all $i$. From Lemma B.3, this implies $\lambda_{\min}(\nabla_x^2 L(x, \lambda)) \leq -\epsilon_H = -\sqrt{\rho\epsilon}$. According to the definition, the algorithm reaches a $\epsilon$-second-order stationary point.

$\square$

## C. Experiment Details and Additional Results

This appendix provides additional experimental results and implementation details that complement the main text. Unless otherwise stated, we follow the experimental setups and hyperparameter choices of the original reconstruction methods, and apply sample splitting as an add-on optimization mechanism without modifying the original objectives or assumptions.

### C.1. Implementation and Computational Details

All experiments are conducted on a single NVIDIA V100 GPU. A single reconstruction run typically takes around 30 minutes, depending on the dataset size and the reconstruction method.

For all splitting-based experiments, the minimum eigenvalue of the splitting matrix is approximated using the Lanczos method (Lanczos, 1950) with a small number of iterations (typically 20). In practice, computing the splitting direction is fast (approximately 2 minutes for 1000 reconstructed samples) and incurs negligible overhead compared to gradient-based optimization. As a result, sample splitting does not constitute a computational bottleneck.

Regarding the splitting procedure, we perform a sample splitting step every 20000–40000 gradient descent iterations, depending on the observed rate of loss decrease. The splitting threshold is set at $\lambda_* = -0.1$, and the total number of split samples is capped at 50% of the current batch to control sample growth. For the splitting step size, we perform a line search along the splitting direction, with the maximum step size capped at 0.01 to ensure numerical stability.

For evaluation, we use L2 distance on MNIST and SSIM on CIFAR-10, as these metrics better correlate with perceptual reconstruction quality for grayscale digits and natural color images respectively, and this choice is consistent with prior reconstruction works.

### C.2. Additional Results for Haim et al. (2022)

We first report extended results for the KKT-based reconstruction method of Haim et al. (2022), which considers homogeneous neural networks trained for binary classification.

**MNIST.** We consider MLPs trained on 500 MNIST samples, following the settings reported in the original paper. Figure 5c presents per-sample L2 distance comparisons before and after sample splitting. Most samples either improve or remain largely unchanged after splitting, indicating that splitting rarely degrades reconstruction quality. Figure 7c visualizes the

top reconstructed samples, selected using the same criterion as Figure 1 in the main text. Qualitatively, improvements from splitting are most visible in background uniformity, digit sharpness, and contrast, rather than large structural changes. Figure 6 further shows optimization trajectories of representative MNIST samples. After splitting events, trajectories often exhibit renewed progress in the reconstruction metric, supporting the interpretation that splitting refines ambiguous reconstructions that have plateaued under gradient-based updates.

**CIFAR-10.** Reconstruction on CIFAR-10 with 500 training samples is more challenging. We find that reconstruction quality is highly sensitive to hyperparameter choices, and a single run often yields only a small number of reasonably reconstructed samples. Figure 5a shows that even in this regime, sample splitting improves reconstruction metrics for many samples. However, as illustrated in Figure 7a, overall visual quality remains limited due to the instability of the baseline method. To better isolate the effect of sample splitting from hyperparameter tuning, we additionally consider smaller training sets of size 100. In this setting, splitting yields modest improvements on relatively well-reconstructed samples, as shown in Figures 5b and 7b.

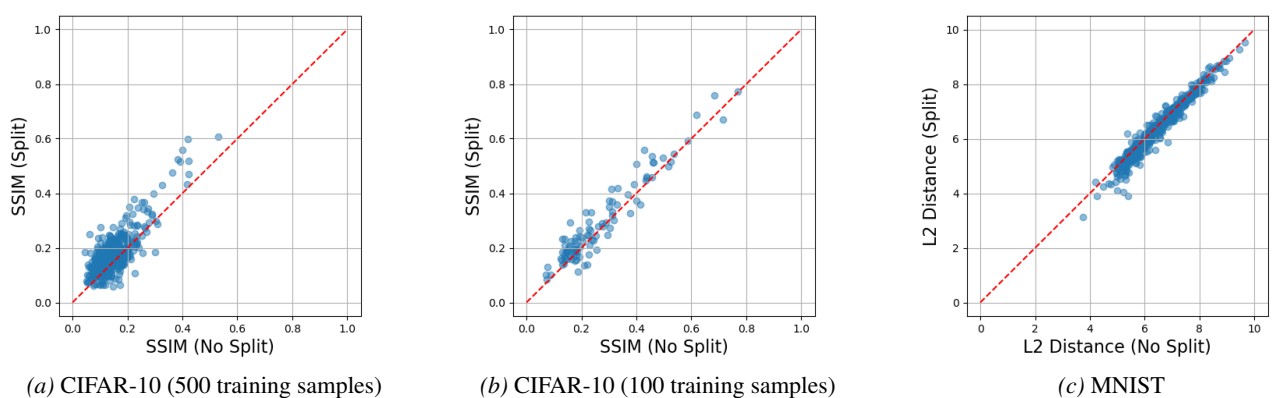

*(a)* CIFAR-10 (500 training samples)      *(b)* CIFAR-10 (100 training samples)      *(c)* MNIST

*Figure 5.* Per-sample metric comparison for CIFAR-10 and MNIST reconstructions using Haim et al. (2022)'s method. Each point corresponds to a training sample, with axes denoting metrics before and after splitting.

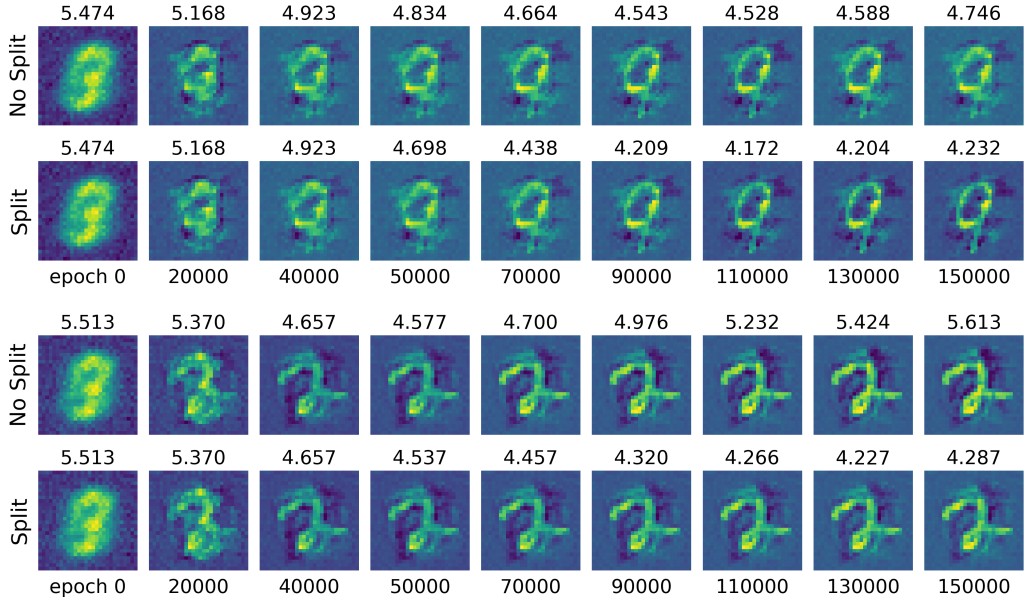

*Figure 6.* Optimization trajectories of representative MNIST samples under Haim et al. (2022)'s method, illustrating metric evolution with and without sample splitting (measured by L2 distance; splitting checked every 40000 epochs).

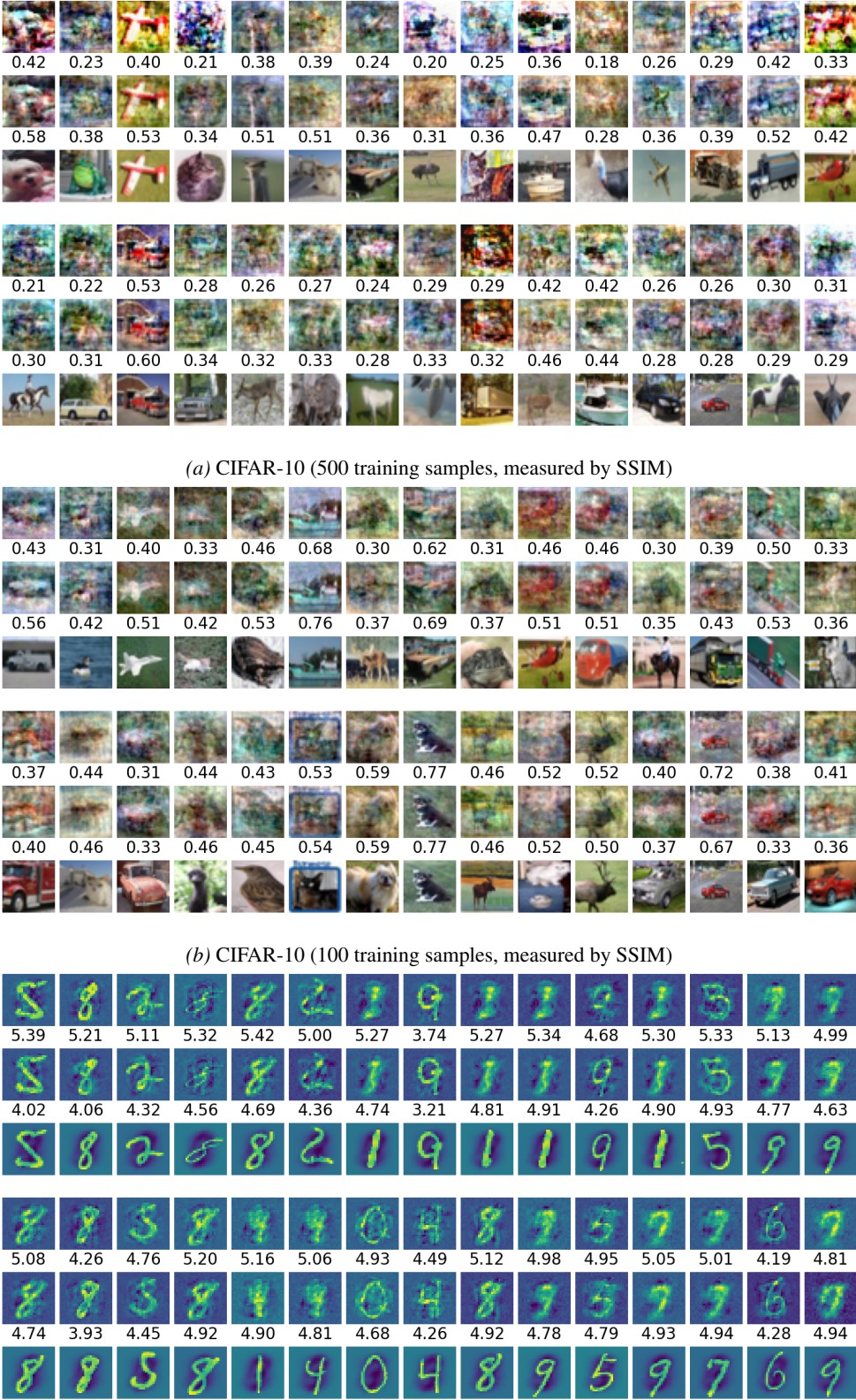

*(a)* CIFAR-10 (500 training samples, measured by SSIM)

*(b)* CIFAR-10 (100 training samples, measured by SSIM)

*(c)* MNIST (500 training samples, measured by L2 distance)

*Figure 7.* Top 30 images reconstructed from MLP trained on 100 images using Haim et al. (2022)'s (rows 1, 4), Haim et al. (2022)'s with sample splitting (rows 2, 5) , and corresponding nearest neighbors from the dataset (rows 3, 6).

## C.3. Additional Results for Buzaglo et al. (2024)

We next report results for the multiclass KKT-based reconstruction method of Buzaglo et al. (2024).

**MNIST.** We reconstruct 10-class MNIST MLP classifiers trained on 500 samples using standard learning rates. Figure 8b reports per-sample L2 comparisons, while Figure 10b shows representative reconstructions. Sample splitting yields improvements for samples that are already reasonably reconstructed. Importantly, splitting seldom degrades overall reconstruction quality, suggesting that it acts as a conservative refinement step. We note that more substantial gains may be achievable with careful hyperparameter tuning.

**CIFAR-10.** For CIFAR-10 with 500 training samples, reconstruction loss fluctuates significantly during optimization, as shown in Figure 9a). This behavior is likely caused by samples repeatedly crossing class margins in the multiclass setting. In this unstable regime, splitting can occasionally produce large improvements for individual samples, although overall performance remains inconsistent.

To better isolate the effect of splitting, we also consider a smaller training set of 100 samples. In this setting, sample splitting improves reconstruction quality even when baseline performance is poor, as shown in Figures 8a, 9b and 10a.

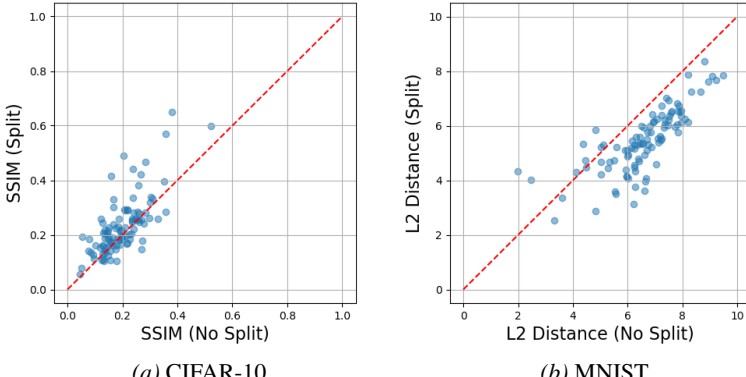

*(a)* CIFAR-10      *(b)* MNIST

*Figure 8.* Per-sample metric comparison for CIFAR-10 and MNIST reconstructions using Buzaglo et al. (2024)'s method.

## C.4. Additional Results for Loo et al. (2024)

Finally, we provide extended results for the NTK-based reconstruction method of Loo et al. (2024). We consider MLPs trained on 100 samples for both MNIST and CIFAR-10 under binary classification.

The original setting sets the initial number of reconstructed samples to $k = 2n$, where $n$ is the (unknown) training set size. Since $n$ is typically unavailable in practice, we further explore the impact of varing $k$. Figure 11 reports loss and metric evolution under different initial reconstruction sizes per class. When $k$ is smaller than the true training set size, the baseline method stagnates, whereas sample splitting enables further progress.

Figures 12 and 13 present per-sample metric comparisons and representative reconstructions, respectively. Figure 14 further shows optimization trajectories of representative samples, illustrating that the improvements triggered by splitting are robust across different initialization size choices.

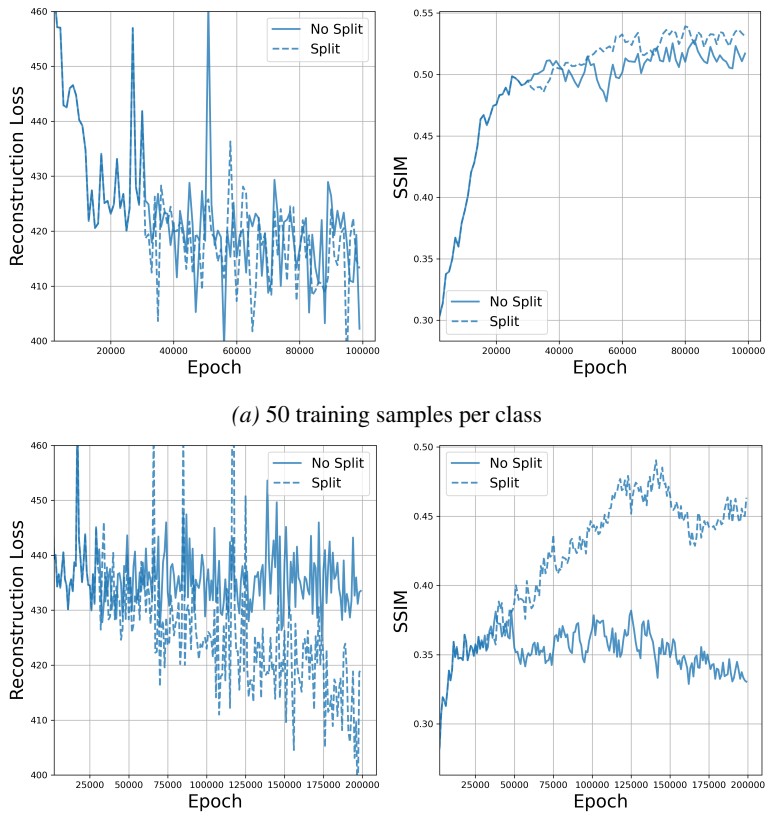

*(a)* 50 training samples per class

*(b)* 10 training samples per class

*Figure 9.* Reconstruction loss and metrics over optimization for CIFAR-10 using Buzaglo et al. (2024)'s method, illustrating instability in the multiclass setting.

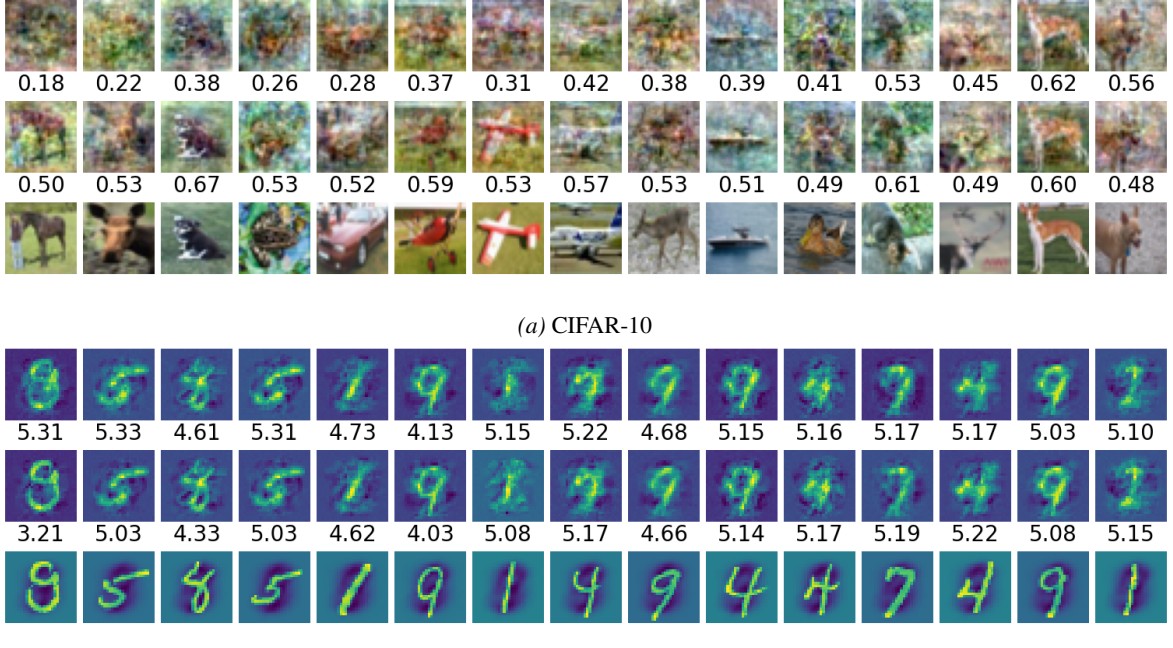

*(a)* CIFAR-10

*(b)* MNIST

*Figure 10.* Top 15 images reconstructed from MLP using Buzaglo et al. (2024)'s (row 1), Buzaglo et al. (2024)'s with sample splitting (row 2) , and corresponding nearest neighbors from the dataset (row 3).

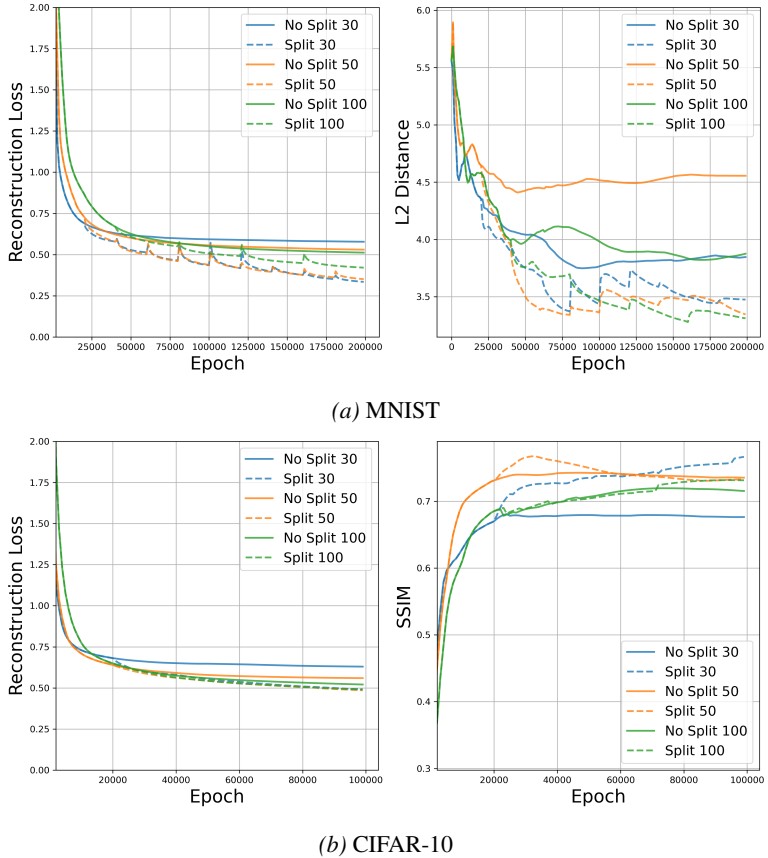

*(a)* MNIST

*(b)* CIFAR-10

*Figure 11.* Loss and metric evolution for MNIST and CIFAR-10 reconstructions using Loo et al. (2024)'s method under different initial reconstruction sizes per class.

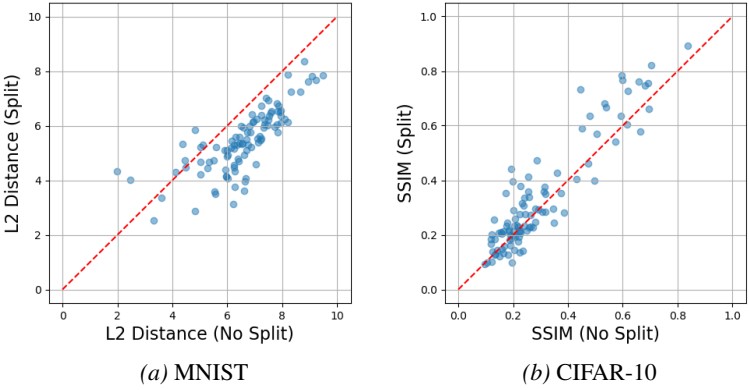

*(a)* MNIST            *(b)* CIFAR-10

*Figure 12.* Per-sample metric comparison for MNIST and CIFAR-10 reconstructions using Loo et al. (2024)'s method with initial reconstruction size of 30 per class.

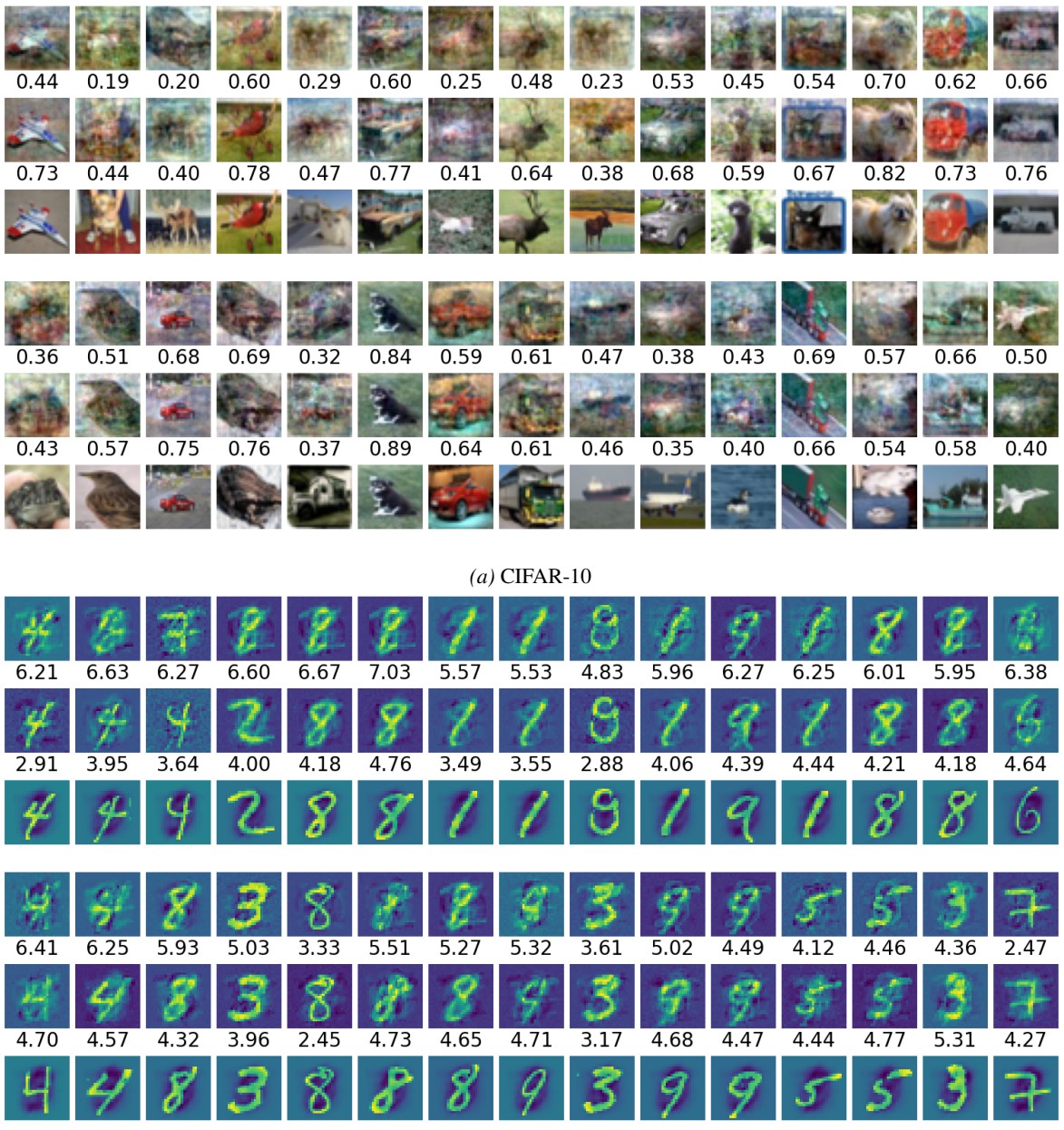

*(a)* CIFAR-10

*(b)* MNIST

*Figure 13.* Top 30 images reconstructed from MLP trained on 100 images using Loo et al. (2024)'s (rows 1, 4), Loo et al. (2024)'s with sample splitting (rows 2, 5) , and corresponding nearest neighbors from the dataset (rows 3, 6).

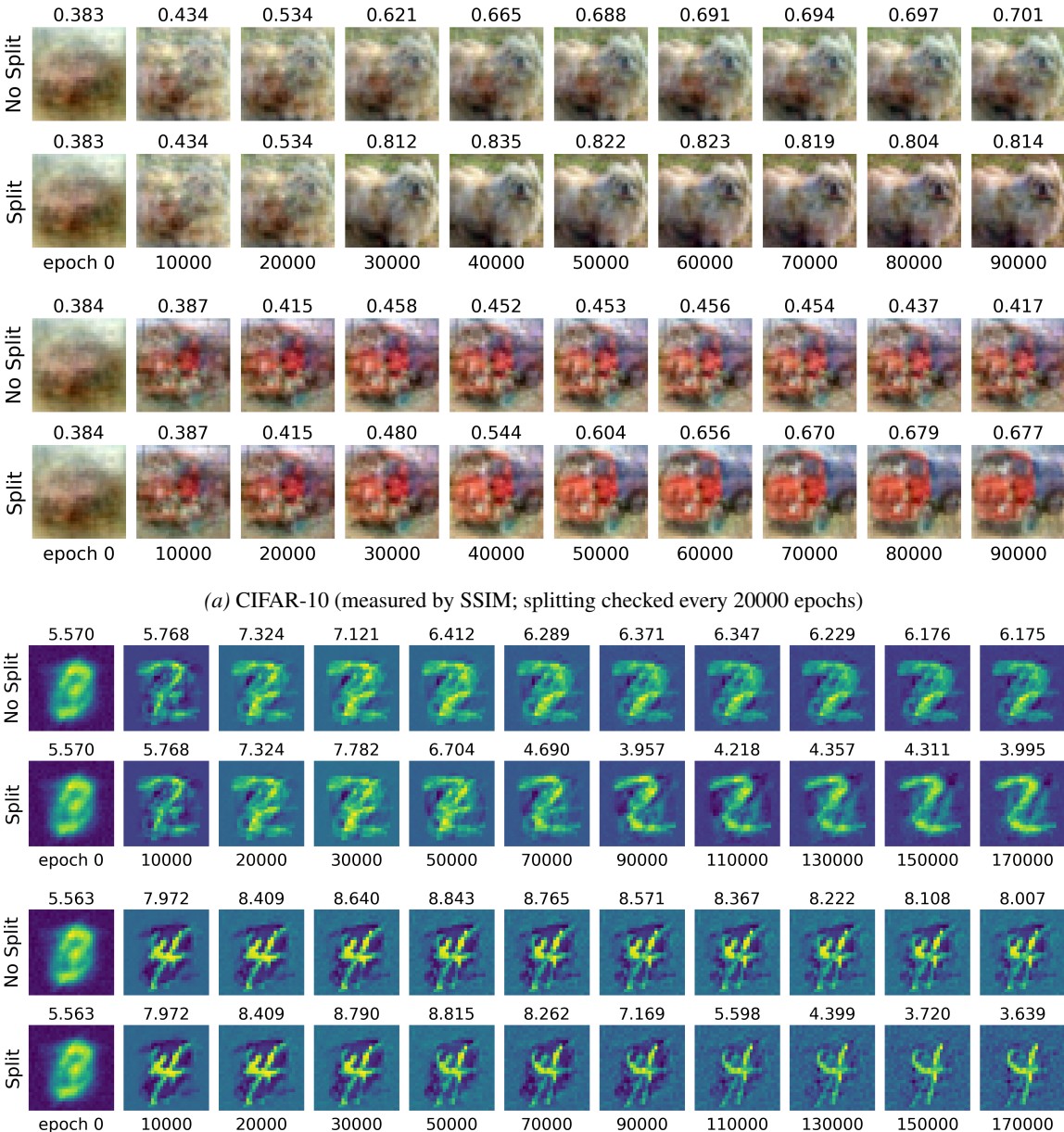

*(a)* CIFAR-10 (measured by SSIM; splitting checked every 20000 epochs)

*(b)* MNIST (measured by L2 distance; splitting checked every 20000 epochs)

*Figure 14.* Optimization trajectories of representative CIFAR-10 and MNIST samples under Loo et al. (2024)'s method, illustrating the effect of sample splitting .

