# OpenReview forum: "Data Reconstruction: Identifiability and Optimization with Sample Splitting"
_ICML.cc/2026/Conference — ICML 2026 regular_

### Official Review · Reviewer_18ku · 2026-03-02

**Soundness:** 4
**Presentation:** 4
**Significance:** 3
**Originality:** 3
**Overall Recommendation:** 5
**Confidence:** 4

**Summary:**

This work studies the recovery of training data assuming access to the trained model in the context of classification. Existing work showed that, when training homogeneous models with the logistic loss with gradient-based methods, the solutions should be close that that of the associated max margin problem. By exploiting its KKT conditions,

The current work studies the scenarios under which using the KKT conditions is sufficient in order to recover the training samples. They show that, with linear or quadratic models, samples cannot be identified. But with polynomial models of degree >= 3, such a recovery is theoretically possible to recover the active samples, i.e., the ones with dual variable lambda_i>0.

However, even when identifiability is possible, recovering the samples requires solving a highly non-convex optimisation problem. Several methods have been developed, which usually set a priori the number of samples to recover, although unknown. The current work proposes to complement such methods with sample-splitting, i.e., . This method is meant to both facilitate the optimisation of the non-convex objective, especially in regions with flat plateaus, and to adaptively control the number of training points to reconstruct. They experimentally demonstrate that sample-splitting yields a lower reconstruction loss during optimisation, and lower distance between the reconstructed samples and the nearest neighbour from the real training data.

**Compliance With Llm Reviewing Policy:**

Affirmed.

**Final Justification:**

After reading the other reviewers rebuttals, I acknowledge the limitations of this work regarding the theoretical scope of the identifiability results.

However, I find that this work nicely extends the work of Haim et al., providing a first step towards theoretically grounding the proposed approach, as well as an improved algorithmic solution for identifying the samples.

I am hence keeping my score as it is.

**Key Questions For Authors:**

1- It is mentioned in the paper that training on a large number of samples makes the sample recovery harder. However, it is not clear, from the analysis, what makes it harder, and where exactly the difficulty arises when the number of training samples grows. A discussion on this aspect would be appreciated.

2- How does the number of samples evolve during splitting. A strength of the algorithm is claimed to be its adaptive sample size. But is there a way to make sure this sample size approaches more or less the true dataset size, or does sample splitting increases the sample size indefinitely ?

3- From Theorem 4.1, it seems that sample identifiability is tightly connected to the interpolation condition, requiring the matrix K to be full rank. It is not clear to me in what case this condition would naturally hold, and what could potentially break it. It might be interesting to analyse the spectrum of this matrix for a toy problem.

4- A visualisation similar to Figure 2 of Haim et al. to better understand which samples are split depending in the Model landscape could be nice to have.

**Limitations:**

Yes, by publishing possible attack algorithms to data privacy, this work attemps to enhance security and privacy of future trained models. It shows in particular that linear and quadratic models demonstrate excellent robustness to this kind of attacks.

**Strengths And Weaknesses:**

Strengths:

1- The paper is well written, the contributions are clearly defined and relevant.

2- The identifiability results from KKT conditions provide strong support of the method, while pinpointing failure cases when using linear or quadratic models.

3- The introduction of sample splitting is theoretically back up, and shows practical improvement of the method on real dataset, both in terms of optimisation performance and sample recovery efficiency.

(4)- In the Experiments section, you mention using a d-1000-1000-1 networks, following Haim et al., but you do not specify the activation function, which seems absolutely critical from the provided analysis !! Hair et al. used a ReLU activation, so I guess you are doing the same, but precisely mentioning it is important. If this is indeed the case here, it is a strength of the paper to demonstrate that the method still works with higher number of hidden layers (although only 3 layers) and a non-polynomial activation function.

Weaknesses:

1- Being based on the algorithm of Haim et al., this work inherits the same limitations, i.e., explaining why would minimising the loss of equation (5) should yield the actual training samples, and the fact that this method using the KKT conditions can only recover the active samples, i.e., the ones on the margin.

2- The identifiability theory only applies to 2-layers networks with polynomial activations. With more layers, the KKT conditions would become much more intricate and much harder to analyse.

3- The empirical evaluation is limited to small fully connected models on toy datasets. At least a discussion about the possible application to other kinds of architectures, e.g., CNNs, would be appreciated. In particular, it might be that the use of weight sharing in such architecture could pose a problem to this kind of method...

---

> ### Author Rebuttal · Authors · 2026-03-31
>
> We thank Reviewer 18ku for the thorough and positive evaluation. We address the
> weaknesses and key questions below.
>
> **[W1] No formal justification for why minimizing Eq. (5) recovers the true data;
> can only recover the active samples.**
> On the lack of formal justification: the core difficulty is that the KKT system
> may admit multiple solutions, making convergence of the optimizer necessary but not
> sufficient for recovering the true data. Our identifiability result (Theorem 4.1)
> addresses this by showing that, under the interpolation condition, the KKT system
> has a **unique** solution equal to the active training samples, transforming what
> was previously a heuristic argument into a conditional formal guarantee.
>
> On recovering only active samples: please see our response to Reviewer SVsH **[W3]**,
> which addresses this point in detail. In brief, sample splitting does not expand
> the recoverable set, but recovers a larger fraction of it under the same threat
> model, providing a stronger and more realistic privacy audit.
>
> **[W2] Theoretical results limited to two-layer networks with polynomial activations.**
> Please see our responses to Reviewer SVsH **[W2.2]** and **[W2.3]**, which address
> polynomial activations and two-layer networks respectively. In brief, we present
> the two-layer polynomial case as the first clean setting for rigorous identifiability analysis, and view it as a fundamental first step toward more general settings.
>
> **[W3] Empirical evaluation limited to small fully connected models.**
> Convolutional networks without bias are homogeneous and satisfy the same KKT
> conditions underlying our analysis. Our focus on fully connected networks follows
> the experimental setup of Haim et al. for a direct and controlled comparison.
> More importantly, our **sample splitting strategy is not tied to the KKT
> formulation**. As discussed in Section 5, it applies to any reconstruction objective
> satisfying mild structural conditions, as demonstrated by the NTK-based experiment of Loo et al. (2024) included in the paper. Extending the method to CNNs is a natural
> next step we are actively pursuing. Regarding implementation details, all experiments use ReLU activations following Haim et al.; we will make this explicit in Section 6.
>
> **[Q1] Why does recovery become harder with more training samples?**
> As $n$ grows, the KKT system involves $O(n)$ dual variables and the optimization
> landscape acquires more saddle points and spurious stationary points. The
> probability of converging to a non-recovering solution therefore increases with
> $n$, consistent with the empirical observation that models trained on fewer samples
> are more vulnerable to reconstruction (Haim et al., 2022; Buzaglo et al., 2024).
>
> **[Q2] How does the sample count evolve during splitting?**
> Sample splitting is not designed to converge to the true dataset size. Its adaptive
> value lies in removing the need to specify $k \geq 2n$ in advance: the algorithm
> begins from a conservative estimate and expands when the optimization landscape
> warrants it.
>
> The table below reports sample size and average minimum eigenvalue of the
> per-sample curvature across splitting steps, for MNIST with 500 training samples
> (initial size 300, splitting period 20,000 epochs).
> The average minimum eigenvalue is negative throughout, indicating that splitting continues to find exploitable descent directions. Its monotonic increase toward zero shows that the algorithm progressively exhausts negative curvature and approaches the stopping criterion. This provides direct mechanistic evidence that splitting steers the optimization toward better-conditioned regions.
>
> | Splitting step | 0 | 1 | 2 | 3 | 4 | 5 |
> |---|---|---|---|---|---|---|
> | Sample size | 300 | 434 | 542 | 670 | 776 | 875 |
> | Avg. min eigenvalue | −0.58 | −0.33 | −0.26 | −0.21 | −0.19 | −0.17 |
>
> **[Q3] Under what conditions does the matrix $K$ satisfy the interpolation
> condition?**
> Please see our response to Reviewer 7cyv **[Q]**, which addresses this in detail.
> In brief, the condition holds generically when the network width exceeds polynomial feature dimension $N$ and the trained neurons are in general position.
>
> **[Q4] Visualization of which samples are split (analogous to Figure 2 of
> Haim et al.).**
> We have prepared the requested visualization (see
> [anonymous link](https://anonymous.4open.science/r/data_rec_rebuttal/)), showing
> reconstruction samples every 500 steps with split children marked. In the 2D toy
> setting, the baseline already recovers all training samples. Nevertheless,
> splitting perturbs points that are stuck near the center, causing their
> children to converge toward training samples more reliably and leaving fewer
> unrecovered central points at the end.
>
> We thank the reviewer for the detailed and constructive questions, which have
> helped us improve the paper.

---

> > ### Author Rebuttal · Reviewer_18ku · 2026-04-02
> >
> > I thank the authors for their detailed rebuttal, which clarified my points of concern.
> >
> > After reading the other reviewers rebuttals, I acknowledge the limitations of this work regarding the theoretical scope of the identifiability results.
> >
> > However, I find that this work nicely extends the work of Haim et al., providing a first step towards theoretically grounding the proposed approach, as well as an improved algorithmic solution for identifying the samples.
> >
> > I am hence keeping my score as it is.

---

### Official Review · Reviewer_SVsH · 2026-03-08

**Soundness:** 3
**Presentation:** 3
**Significance:** 2
**Originality:** 2
**Overall Recommendation:** 4
**Confidence:** 3

**Summary:**

This paper studies the problem of dataset reconstructure from KKT condition. The paper has two main first results: (1) the authors proposed an identifibility condition for dataset reconstructure on two layer networks with polynomial activation functions. (2) the authors proposed a sample splitting scheme for recover to solve the problem of unknow size of dataset and potential convergence to the saddle points. The authors also implements the algorithms on practical datasets.

**Compliance With Llm Reviewing Policy:**

Affirmed.

**Final Justification:**

In my original evaluation, I found the paper technically interesting; however, I was not fully convinced that the studied setting is sufficiently interesting, particularly regarding the usefulness and scope of the identifiability results, as well as the significance of designing new recovery algorithms.

After reading the rebuttal, I acknowledge that this work provides the first formal characterization of identifiability in this context, and I appreciate the authors’ motivation for studying a clean and simplified setting. While I personally still feel that identifiability may not be central to the data privacy problem, and that the scope of the setting is somewhat restrictive, I respect that different researchers may place different value on these aspects.

Regarding the usefulness of sample splitting, I appreciate the authors’ explanation. On the other hand, I find the argument somewhat qualitative. For example, the response that “without sufficiently strong optimization, one cannot distinguish true non-recoverability from optimizer failure” is intuitively reasonable, but it remains unclear to me whether this translates into a meaningful difference in terms of algorithmic hardness between approaches with and without sample splitting. However, I agree that these questions are likely beyond the scope of the current paper, and I will not pursue them further here.

In conclusion, I find the paper technically interesting, and the problem studied is potentially meaningful. Thus, I am willing to increase my score to 4.

**Key Questions For Authors:**

- The  identitifiability conditions tells when the KKT system has a unique solution. I wonder is it possible that the system does not have a unique solution, but the recovery algorithm still converge to the correct solution that corresponds to the datapoints?

- The  identitifiability conditions require $rank(K) = N,$ which implies the width of the network $m \geq N \approx d^{\alpha -1}.$ I wonder when $m$ is smaller, is it still possible to get some results from  the recovery?

**Limitations:**

Yes

**Strengths And Weaknesses:**

### Strengths:
1. For the identifiability of two-layer polynomial networks, the authors reduce the KKT condition to the interpolation linear system, and the identifiability of the network depends on the uniqueness of the solution interpolation linear system and the degree of the tensor $\mathcal T,$ which I think it's an interesting finding theoretically.

2. The proposed sample splitting method is simple but well-motivated in my opinion, and it seems to give better recovery given the experimental results in section 6.

### Weaknesses:

1. While I acknowledge the technical contribution of the proposed identifiability results, high-levelly speaking, I feel this results is a bit incremental. In particular, previous papers such as (Haim et.al 2022) already gave intuitions that one can recover datasets from the KKT condition and worked well in practice, and the identifiability results in this paper give a condition on when the KKT system yield a unique solution under a limited setting, which is a complementary to previous works. Could the authors elaborate more on the usefulness of the identifiability results, and how it provides new intutions compare to previous paper?

2. The scope of identifiability results seems to be restrictive.

    (1) The general setting is data recovery from KKT point, which itself is restrictive in my opinion. In particular, if I understand correctly, the weights converge to KKT points only when training with loss of exponential tail on linear sparable data, and without weight decay, which is not always the case in practice.

    (2) The identifiability results only apply to polynomial activations. In particular, the results heavily relies on the fact that $f(w)$ is in a finite-dimensional linear space, which is true for polynomial activations, but not for non-polynomial but homogeneous activations such as ReLU if I understand correctly. While I appreciate the discussions in section 4.3 on non-homogeneous polynomial networks, I wonder if the authors could elaborate more on the application of the theory on the non-polynomial activations?

   (3) Also, the identifiability results is only two-layer network, and it seems not obvious to extend to deeper networks.


3. While I acknowledge that the proposed sample splitting methods could improve the recovery quality and the nice theorecial motivation behind, I'm not sure the significance of designing new recovery algorithms that have better recovery quality. From a data privacy point of view, it seems identifying which data point is recoverable is key, as then we could try to design methods to protect the privacy of those point. The sample splitting doesn't seems to recovery new type of points, i.e. it still can only recovery the points on the margin. Thus, could the authors elaborate more on the usefulness of the proposed methods?

---

> ### Author Rebuttal · Authors · 2026-03-31
>
> We thank Reviewer SVsH for the detailed feedback. We address each concern below.
>
> **[W1] Incrementality relative to Haim et al. (2022).**
> We respectfully disagree with the reviewer on this point. Haim et al. demonstrated that training data satisfy the KKT conditions and **empirically** showed the possibility to recover samples in some cases. However, their contribution is purely empirical and does not resolve the two central questions behind KKT-based reconstruction (we have also emphasized in the introduction):
> (i) when does the KKT system uniquely identify the true training data, and (ii) is there a computationally efficient procedure that can reliably recover them? These
> questions are genuinely open and challenging due to the nature of non-convexity in the considered problem. In fact, subsequent work (Rafael et al. 2025) proved the original KKT-based objective suffers from exponentially many local minima,
> partially confirming that both questions were far from settled by prior empirical
> demonstrations.
>
> Our work makes the first progress on both fronts. On identifiability, we provide
> explicit sufficient conditions under which the trained model uniquely determines the
> active training samples, and characterize failure cases such as linear and quadratic
> settings. On the algorithmic side, we introduce sample splitting as a principled
> optimization mechanism. We therefore view our results as complementary to, rather
> than incremental over, Haim et al.
>
> [Rafael et al. 2025] No Prior, No Leakage: Revisiting Reconstruction Attacks in Trained Neural Networks.
>
> **[W2] The scope of identifiability results seems to be restrictive.**
> We have to clarify that our work is the **first formal identifiability characterization**, so we choose a clean regime in our analysis. We address each sub-point in the following.
>
> **[W2.1] KKT convergence is restrictive.**
> The KKT system is exactly the core object of the reconstruction framework of Haim
> et al., so understanding identifiability here addresses the central missing
> theoretical question behind that attack. In this sense, our result is deliberately scoped to the first clean setting where one can rigorously ask whether the trained network uniquely determines the data. Both their experiments and ours work well
> beyond the strict theory, suggesting that KKT geometry remains informative more
> broadly, even if a formal extension has not yet been proved.
>
> **[W2.2] Identifiability restricted to polynomial activations.**
> Please see our response to Reviewer iVVF **[W1]**, which addresses this point in
> detail. In brief, we view the polynomial case as a necessary first step, and the
> experiments on ReLU networks already in the paper suggest the theoretical intuition
> carries over in practice.
>
> **[W2.3] Results limited to two-layer networks.**
> We agree that extending the theory to deeper networks is harder because the KKT
> equations become much more coupled and nonlinear across layers. We present the
> two-layer case as the first clean setting for rigorous identifiability analysis and it is also the minimal nonlinear setting. Since identifiability already arises here, we view this as a fundamental first step toward deeper networks.
>
> **[W3] Sample splitting recovers only margin points; no new type of point is recovered.**
> Identifying which samples are structurally recoverable is only part of the picture.
> Without sufficiently strong optimization, one cannot distinguish true
> non-recoverability from optimizer failure. Sample splitting does not change the
> fact that KKT-based attacks target active samples, but it recovers a larger
> fraction of that vulnerable set under the same threat model, showing that part
> of the under-recovery in prior work is due to optimization barriers rather than
> lack of information. Privacy evaluations should therefore be based on the strongest
> practical attack, and our method provides a stronger and more realistic privacy
> audit.
>
> **[Q1] Could the system have non-unique solutions yet the algorithm still converge
> to the correct data?**
> Yes, this is possible in principle. In the non-unique case, the algorithm may
> converge to a solution consistent with the KKT conditions, but there is no
> guarantee it corresponds to the original training data. This is precisely the
> motivation for our identifiability result: it certifies that convergence to a KKT
> solution implies recovery of the true data. Without identifiability, convergence
> is necessary but not sufficient.
>
> **[Q2] Can partial results be obtained when $m$ is below the identifiability
> threshold?**
> This is an interesting open question. Theorem 4.1 gives sufficient, not necessary,
> conditions. As discussed in the last section, when the full-rank threshold fails, partial or approximate recovery based on dominant spectral components may still be possible, which we identify as a direction for future work.
>
> We thank the reviewer again. If our responses above have addressed your concerns, we would truly appreciate a re-evaluation accordingly.

---

> > ### Author Rebuttal · Reviewer_SVsH · 2026-04-02
> >
> > I thank the authors for addressing my questions and concerns in detail.
> >
> > In my original evaluation, I found the paper technically interesting; however, I was not fully convinced that the studied setting is sufficiently interesting, particularly regarding the usefulness and scope of the identifiability results, as well as the significance of designing new recovery algorithms.
> >
> > After reading the rebuttal, I acknowledge that this work provides the first formal characterization of identifiability in this context, and I appreciate the authors’ motivation for studying a clean and simplified setting. While I personally still feel that identifiability may not be central to the data privacy problem, and that the scope of the setting is somewhat restrictive, I respect that different researchers may place different value on these aspects.
> >
> > Regarding the usefulness of sample splitting, I appreciate the authors’ explanation. On the other hand, I find the argument somewhat qualitative. For example, the response that “without sufficiently strong optimization, one cannot distinguish true non-recoverability from optimizer failure” is intuitively reasonable, but it remains unclear to me whether this translates into a meaningful difference in terms of algorithmic hardness between approaches with and without sample splitting. However, I agree that these questions are likely beyond the scope of the current paper, and I will not pursue them further here.
> >
> > In conclusion, I find the paper technically interesting, and the problem studied is potentially meaningful. Thus, I am willing to increase my score to 4.

---

### Official Review · Reviewer_7cyv · 2026-03-12

**Soundness:** 3
**Presentation:** 4
**Significance:** 4
**Originality:** 4
**Overall Recommendation:** 4
**Confidence:** 4

**Summary:**

This paper considers the both the theoretical and algorithmic sides for dataset reconstruction. Theoretically, it establishes the results when KKT-based reconstruction is identifiable by establishing sufficient conditions for unique recovery in two-layer networks with polynomial activations. Algorithmically, it introduces a stragety called: sample splitting, a curvature-based refinement strategy that creates new descent directions. The numerical experiments show that the proposed optimization helps escape poor stationary points and improve empirical reconstruction performance.

**Compliance With Llm Reviewing Policy:**

Affirmed.

**Final Justification:**

I will keep my score since it is already positive.

**Key Questions For Authors:**

Regarding the weakness 1, I have the following question:
 Since this grows combinatorially with $d$ and $\alpha$, could the authors clarify in what regimes this condition is expected to hold for trained neurons $\{W_j\}$?

**Limitations:**

yes

**Strengths And Weaknesses:**

Strengths:

1. The paper studies an important problem and provides two aspects of contributions: identifiability and optimization. On the theory side, the paper gives sufficient conditions for unique KKT-based recovery in two-layer networks with polynomial activations. On the algorithmic side, it proposes a curvature-based sample splitting strategy that is applicable beyond the specific KKT formulation.

2. The paper is generally well structured, and the empirical section covers multiple reconstruction pipelines.

Weaknesses:
1. The assumptions in Theorem 4.1  could be strict. In particular, the neuron-rank condition $\mathrm{rank}(K)=N$ seems restrictive, where   $N=\binom{d+\alpha-2}{\alpha-1}$ is the dimension of the degree-$(\alpha-1)$ homogeneous polynomial space. Since this quantity grows combinatorially with $d$ and $\alpha$, the theorem may apply only in certain small regimes.

2. The improvement gained by splitting on CIFAR-10 is not obvious enough, especially compared with the result on
MNIST(see Figure 2) while the improvement on MNIST is minor with 500 training samples(see Figure 5).

Minors:
1. Section 3 raises the central question of why minimizing the reconstruction objective should recover the true data, but the transition from this discussion to the later identifiability results is not fully convincing. In particular, the conceptual role of Section 3 and its connection to the later theory could be clarified.

2.    There is a noticeable citation / referencing issue in Appendix~C.3, where unresolved placeholders (e.g., ``Figures ??'') remain in the manuscript.

---

> ### Author Rebuttal · Authors · 2026-03-31
>
> We thank Reviewer 7cyv for the constructive feedback and positive assessment.
> We address the weaknesses, questions, and minor comments below.
>
> **[W1] Neuron-rank condition grows combinatorially and may restrict the theorem to small regimes.**
> We agree that the interpolation condition in Theorem 4.1 can be restrictive, since the relevant polynomial feature dimension grows combinatorially with input dimension
> and activation degree. Our goal is therefore not to give a complete characterization of practical reconstruction, but to provide a first formal identifiability result
> with explicit sufficient conditions. This dependence is tied to the exact-recovery analysis rather than being a proof artifact.
>
> Moreover, we believe the main value of the theorem is to connect reconstruction to a tensor structure. Through spectral analysis, future work on partial or approximate
> recovery based on dominant spectral components may relax the full-rank requirement. More broadly, if $m$ is viewed as the number of parameters in this minimal analyzable
> setting, the condition is consistent with the overparameterized regime of modern deep networks.
>
> **[W2] Improvement on CIFAR-10 is not obvious; improvement on MNIST with 500 samples is minor.**
> We thank the reviewer for this observation. The trend is consistent with theoretical expectations: KKT-based reconstruction is harder for more complex datasets and larger training sets. CIFAR-10 produces a more ill-conditioned KKT system than MNIST, and larger training sets increase the dimensionality of the optimization landscape. Smaller absolute gains in harder regimes are therefore expected, and are already visible in the baseline performance itself.
>
> In addition, the improvements are more significant than they appear visually. On CIFAR-10 (Figure 2), average SSIM increases from 0.297 to 0.325, a relative
> gain of 9.4%. On MNIST with 500 training samples (Figure 5), average L2 distance decreases from 6.51 to 6.30. Notably, in the latter case, the no-split baseline uses the best hyperparameters reported in Haim et al., obtained via extensive search. The fact that sample splitting yields consistent improvement on top of an already
> well-tuned baseline is, we believe, a meaningful result. With a search tuned for the split setting, more pronounced gains are achievable; see an example at [anonymous link](https://anonymous.4open.science/r/data_rec_rebuttal/).
>
> **[Q] In what regimes is the neuron-rank condition expected to hold for trained neurons?**
> The condition in $m$ holds whenever the network is sufficiently wide relative to the polynomial degree $\alpha$ and input dimension $d$. As for when the condition in $m$ holds, we view it as a generic non-degeneracy assumption: once the width is compatible with the relevant feature dimension, full rank is natural when trained neurons are in general position. It can fail under strong weight symmetries, repeated neurons, or near-collinearity, but these are non-generic configurations that do not arise in standard training.
>
> **[Minor 1] Section 3 to identifiability transition is not fully convincing.**
> Section 3 motivates *why* we study the KKT system as a route to data reconstruction. However, the KKT system alone does not guarantee a unique solution, which is necessary for reliable reconstruction. Section 4 then characterizes *when* that uniqueness holds. The identifiability result is the formal realization of the intuition in Section 3. We will add a bridging sentence at the end of Section 3 to make this transition explicit.
>
> **[Minor 2] Unresolved "Figures ??" placeholders in Appendix C.3.**
> We apologize for this oversight. These references will be corrected in the revised manuscript.
>
> We thank the reviewer for the careful reading and constructive suggestions. If our responses above have addressed your concerns, we would truly appreciate your strong support.

---

> > ### Author Rebuttal · Reviewer_7cyv · 2026-04-02
> >
> > Thank you for the rebuttal. The responses addressed most of my concerns.
> >
> > In particular, the clarification on the neuron-rank condition helps better position Theorem 4.1 as a first formal identifiability result under explicit sufficient conditions. The additional quantitative explanation of the sample-splitting gains is also helpful, although the improvements in harder regimes still appear somewhat modest.
> >
> > Overall, the rebuttal increases my confidence in the paper, and I am comfortable maintaining my weak accept recommendation.

---

### Official Review · Reviewer_iVVF · 2026-03-13

**Soundness:** 3
**Presentation:** 3
**Significance:** 2
**Originality:** 2
**Overall Recommendation:** 4
**Confidence:** 2

**Summary:**

The paper is concerned with the problem of reconstructing the dataset used to train a neural network, given access to the weights of the  network.  Ability to reconstruct the dataset could potentially raise privacy concerns and might also be related to questions related to memorization and generalization.

The theoretical portion of the paper focuses on 2-layer neural networks with polynomial activation functions. For this type of networks, the authors show that for polynomials of degree $\alpha\geq 3$ reconstruction is, in principle, possible under some non-degeneracy assumptions of the training data. For the computational question i.e., recovering the dataset algorithmically,  the authors also propose a splitting strategy and show that the algorithm converges to a second order stationary point. The splitting strategy is experimentally evaluated and appears to improve reconstruction compared to previous methods.

**Compliance With Llm Reviewing Policy:**

Affirmed.

**Final Justification:**

I think that the paper is overall interesting and given that the authors address relationships to known the literature I am increasing my score to weak accept.

**Key Questions For Authors:**

It seems that the condition on $x_i$ in Theorem 4.1, namely that the vectors $x_i$ are linearly independent, is essentially the same condition that appears in the analysis of Jennrich's algorithm, which also guarantees uniqueness of the tensor decomposition (see, for example, Chapter 3 of "Algorithmic Aspects of Machine Learning" by Ankur Moitra). The argument in the appendix appeared quite similar to the standard analysis used for Jennrich's algorithm. It is possible that I am missing a subtle point and that what you need is stronger/different. In any case, I think it would be helpful to clarify the relationship to existing uniqueness results for tensor decompositions, and to add an explicit citation and short discussion.

**Limitations:**

yes

**Strengths And Weaknesses:**

Strengths:
The paper is well written and the proofs are presented in a rigorous and clear manner. The question of reconstructing the training dataset from the weights of a neural network is interesting and potentially impactful, especially for understanding issues related to privacy and memorization. Studying this question from a principled theoretical perspective and proving formal guarantees is an exciting  direction.

Weaknesses:
The theoretical contribution is somewhat limited by the fact that the activation functions are restricted to polynomials. While polynomial activations have often been used as a stepping stone toward a theoretical understanding of deep learning, it would be significantly more compelling to obtain identifiability results for more commonly used activation functions, such as ReLU. Furthermore, the main results are not entirely surprising given uniqueness results for higher-order tensor decompositions, in particular for tensors of order greater than three (see also question below).

---

> ### Author Rebuttal · Authors · 2026-03-31
>
> We thank Reviewer iVVF for the careful reading and for raising the important connection to the tensor decomposition literature.
>
> **[W1] Identifiability restricted to polynomial activations.**
> The polynomial setting is adopted because $\phi_\alpha(\mathbf{x})$ lies in a finite-dimensional linear space, enabling a clean algebraic analysis via the interpolation system. Our goal is not to claim a general reconstruction theorem for arbitrary architectures, but rather to provide a **first formal identifiability characterization** within the trained neural network reconstruction framework of Haim et al. (2022). Within this scope, our work contributes two concrete insights: (1) it identifies failure cases for linear and quadratic models, and (2) it gives explicit sufficient conditions under which the KKT system uniquely determines the active training samples. More broadly, polynomials are dense in continuous functions on compact domains, so we view the polynomial case as a natural first step toward understanding more general activations.
>
> **[W2] Results not surprising given Jennrich's algorithm; appendix proof appears similar.**
> We thank the reviewer for this insightful observation. We agree that Step 2 of our proof uses a Jennrich-style argument, but we need to emphasize that our main contribution here is to analyze an open problem in data reconstruction tasks.
>
> More precisely, our proof has two conceptually distinct steps. First, we show that the KKT equations uniquely determine the underlying symmetric tensor under the interpolation conditions. Second, once this tensor is identified, we use a Jennrich-style decomposition argument to recover the active components. Even in Step 2, we do more than invoke Jennrich on a given tensor: we also verify that the tensor objects required for the decomposition are computable in the data reconstruction setting. Therefore, we do not claim a new tensor-decomposition uniqueness theorem. Rather, our contribution is to reduce the identifiability in a data reconstruction task to a uniqueness statement in tensor decomposition. Applying an existing mathematical tool (Jennrich's algorithm) makes our analysis direct and simple, which is an advantage instead of a drawback of the work. As the reviewer suggested, we will clarify the connection with Jennrich's algorithm explicitly in the revision by adding the appropriate citation and discussion.
>
> **[Q] Clarify relationship to Jennrich; add explicit citation.**
> Please see the explanation in **[W2]** above.
>
> We hope this clarification addresses the reviewer's concern about novelty, and we would truly appreciate a re-evaluation if our responses have resolved the main issues.

---

> > ### Author Rebuttal · Reviewer_iVVF · 2026-04-02
> >
> > I raise my score but I think it is quite important to address the relationship to Jennrich's algorithm. I think this was a serious omission by the authors.

---

### Decision · Program_Chairs · 2026-04-30

**Decision:**

Accept (regular)

**Comment:**

This paper
1. provides, for a 2 layer NN with polynomial activation functions with degree $\geq 3$, a first formal characterization of sufficient conditions on the KKT system to make it possible to reconstruct the training set
2. proposes in practice to add in a concrete algorithmic reconstruction approach a splitting strategy to escape saddle points
3. proves that the algorithm converges to a 2nd order stationary point
4. presents nice experimental results

All reviewers were satisfied by the responses provided to their questions and comments. It is clear that the work has sufficient novelty, could offer an interesting first step towards stronger characterizations of when similar algorithms recover all or part of the training dataset.

I therefore clearly recommend the paper for acceptance.

Recommandations to the authors to improve the manuscript:

- Following the exchange with reviewer iVVF, the authors proposed to "clarify the connection with Jennrich's algorithm explicitly in the revision by adding the appropriate citation and discussion". This is indeed an important connection to make explicit, so the authors are strongly encouraged to take care of this point.

- Include comments on the neuron-rank condition (question of reviewer 7cyv) in a discussion in the paper.

- The authors are encouraged to take into account the constructive questions and comments of all reviewers in particular on the relation with related work to clarify in the final version the unique contribution of this work what are the questions that this work addresses and which is not addressed in related work.